# Sequential Information Bottleneck Fusion: Towards Robust and Generalizable Multi-Modal Brain Tumor Segmentation

**Tianyi Liu**[1], **Xi Yang**[1], **Wei Wang**[1], **Anh Nguyen**[2], **Haochuan Jiang**[2,†], **Kaizhu Huang**[3]
[1]Xi'an Jiaotong-Liverpool University    [2]University of Liverpool    [3]Duke Kunshan University
[†]Corresponding author: h.jiang@xjtlu.edu.cn

## Abstract

Brain tumor segmentation in multi-modal MRIs poses significant challenges when one or more modalities are missing. Recent approaches commonly employ *parallel* fusion strategies; however, these methods often risk losing crucial shared information across modalities, which can degrade segmentation performance. In this paper, we advocate leveraging sequential information bottleneck fusion to effectively preserve shared information across modalities. From an information-theoretic perspective, sequential fusion not only produces more robust fused representations in missing-data scenarios but also achieves a tighter generalization upper bound compared to parallel fusion approaches. Building on this principle, we propose the Sequential Multi-modal Segmentation Network (SMSN), which integrates an Information-Bottleneck Fusion Module (IBFM). The IBFM sequentially extracts modality-common features while reconstructing modality-specific features through a dedicated feature extraction module. Extensive experiments on the BRATS18 and BRATS20 glioma datasets demonstrate that SMSN consistently outperforms traditional parallel fusion-based baselines, achieving exceptional robustness in diverse missing-modality settings. Furthermore, SMSN exhibits superior cross-domain generalization, as evidenced by its ability to transfer a trained model from BRATS20 to a brain metastasis dataset without fine-tuning.

## 1 Introduction

Brain tumors pose a significant threat to human health. Accurate brain tumor segmentation plays a pivotal role in treatment planning by precisely identifying tumor boundaries in multi-modal Magnetic Resonance Imaging (MRI) modalities (Yan et al., 2020; Bakas et al., 2017). It is essential to provide all the relevant modalities to achieve proper segmentation performance (Shaker et al., 2024). However, in real-world clinical practice, it is commonly seen that one or more MRI modalities are unavailable due to various practical defects (Liu et al., 2023; Tran et al., 2017).

$$(X_1, X_2, \ldots, X_M) \xrightarrow{\text{combine}} X \xrightarrow{f(\cdot)} Z \quad \Big| \quad X_1, X_2 \xrightarrow{f_1(\cdot)} Z_1, X_3 \xrightarrow{f_2(\cdot)} Z_2, X_4 \cdots \xrightarrow{f_t(\cdot)} Z_t$$
$$\underbrace{\qquad\qquad}_{\text{Parallel Fusion}} \qquad\qquad \underbrace{\qquad\qquad}_{\text{Sequential Fusion}}$$

Table 1: Illustration of parallel and sequential fusion strategies. $X$ denotes the modality input, $f(\cdot)$ or $f_t(\cdot)$ are fusion functions, and $Z$ is the latent representation.

Recently, the missing-modality challenge has been widely addressed through unified multi-modal feature learning, where joint fusion representations are constructed from available modalities (Zhang et al., 2022; Shi et al., 2023). These methods aim to disentangle modality-common and modality-specific information (Zhao et al., 2023; Liu et al., 2025a). Typically, modalities are fused **in parallel** via simple concatenation (Zhang et al., 2022) or attention mechanisms (Shi et al., 2023; Wang et al., 2025), where all modalities are combined and mapped into a shared latent representation, as shown in Table 1. However, when certain modalities are absent, the common information may not be adequately preserved, resulting in degraded segmentation performance, as illustrated in Figure 1.

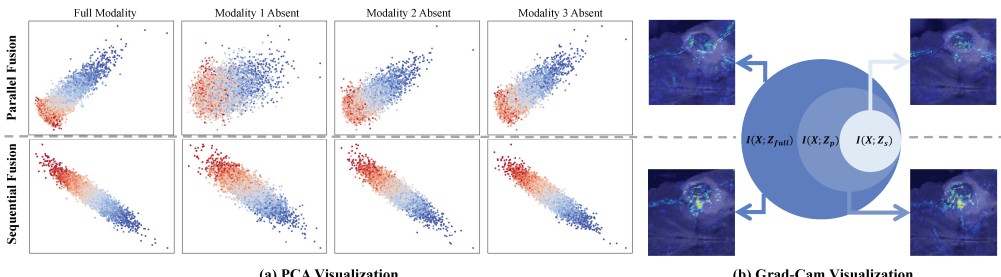

Figure 1: Comparison of parallel fusion (top) and sequential fusion (bottom). (a) PCA Visualization. Large distribution variation when an informative modality is missing indicates weaker robustness, while compact and consistent embeddings with preserved color gradients indicate better information preservation. Modalities 1-3 contain decreasing task-relevant information. Different points represent different samples. (b) Grad-cam Visualization. Sequential fusion ($I(X; Z_s)$) preserves more information when some modalities are missing, while parallel fusion ($I(X; Z_p)$) preserves less.

Given the aforementioned disadvantage in parallel fusion paradigms, we advocate that modalities should be considered by the **sequential information bottleneck fusion**, where modalities are incorporated step by step through recursive updates of the latent state, as shown in Table 1. It could properly preserve more modality commonality. As illustrated in Figure 1, when some modalities are missing, it will not heavily influence the mutual information between the joint inputs and latent fusion distribution. Importantly, a series of theoretical analysis (see Sec. 3) shows that our sequential strategy offers a Lipschitz-continuous lower bound. Compared with parallel fusion, it certifies more stable fused representations in missing-modality scenarios. On the empirical front, loss landscapes (Figure 4) further reveal smoother and flatter optimization surfaces, thus supporting this advantage. Additionally, our proposed method can yield a provably tighter generalization upper bound, vs. parallel fusion, thus confirming again the advantages of our sequential fusion strategy.

Building on these findings, in this paper, we develop the **Sequential Multi-modal Segmentation Network (SMSN)**. Inspired by the ITHP (Xiao et al., 2024), we introduce a Sequential Fusion Module based on the Information Bottleneck objective (IBFM) in the SMSN to facilitate the extraction of modality-common information regardless of whether modalities are fully available or partially missing. To further disentangle modality-specific representations, we employ a Transformer-based Feature Extraction Module, coupled with a series of task-specific losses (Ronneberger et al., 2015).

We validate SMSN on two benchmark datasets, BRATS18 and BRATS20 (Menze et al., 2014), where improved robustness over state-of-the-art fusion-based baselines can be observed. We further generalize a trained SMSN on the BRATS20 glioma dataset directly to an unseen brain metastases dataset (Ramakrishnan et al., 2024) without any fine-tuning performed, where superior generalization performance can be observed comparing with other baselines. Major contributions of this paper are summarized as follows:

- We propose a novel Sequential Multi-modal Segmentation Network (SMSN) to decompose the modality-common information with the information bottleneck based fusion module (IBFM) in missing modality segmentation task.
- We theoretically evidence the generalization and robustness of the sequential IBMF, demonstrating its superiority to parallel fusion.
- We empirically verify the robustness and generalization of the proposed SMSN on various brain tumor segmentation datasets.

## 2 RELATED WORK

### 2.1 MULTI-MODAL BRAIN TUMOR SEGMENTATION

Multi-modal MRI provides complementary information for brain tumor segmentation (Shaker et al., 2024; Fang et al., 2025), but handling missing modalities remains a major challenge in clinical prac-

tice (Liu et al., 2025b). Non-fusion methods, such as modality reconstruction (Liu et al., 2023) or shared-specific feature modeling (Wang et al., 2023), have been explored, but their computational overhead and limited scalability hinder wide adoption. Consequently, most existing works are fusion-based, typically following a parallel fusion paradigm, for example, mmFormer (Zhang et al., 2022), M$^2$FTrans (Shi et al., 2023), MMMViT Qiu et al. (2024), and IMS$^2$Trans Zhang et al. (2024), through concatenation or attention. However, parallel fusion may fail to preserve modality-common information when some modalities are missing, leading to performance degradation. In this paper, we propose a sequential fusion framework that explicitly addresses this limitation and enhances robustness under missing-modality scenarios.

## 2.2 MULTI-MODAL INFORMATION BOTTLENECK

The Information Bottleneck (IB) models the trade-off between information compression and task relevance. It has been proven to be effective in speech and text classification tasks (Slonim & Tishby, 2000; Tishby & Zaslavsky, 2015; Fang et al., 2026). Recently, IB-based techniques have witnessed achievement for improving representation quality for multi-modal learning. The ITHP (Xiao et al., 2024), for example, employs an IB-driven hierarchical framework to distill information from auxiliary modalities into a compact, task-relevant representation. Building on these insights, we adopt the IB to address missing modality challenges in multi-modal learning.

## 3 THEORETIC MOTIVATION

**Theorem 1** (Optimality of the Information Bottleneck Representation for Multi-Modal Inputs). *Let $Y$ be a target variable and $\{X_1, X_2, \ldots, X_N\}$ be $N$ input modalities that are jointly distributed with $Y$. Assume that the modalities can be modeled jointly as a composite random variable $X = (X_1, X_2, \ldots, X_N)$, with a joint distribution $p(x_1, \ldots, x_N, y)$. Let $Z$ be a stochastic representation of $X$, learned via an encoder $p(z|x_1, \ldots, x_N)$, by optimizing the Information Bottleneck (IB) objective:*

$$Z^* = \arg \max_{p(z|x)} \left[ I(Z; Y) - \beta I(X; Z) \right], \tag{1}$$

*where $\beta > 0$ controls the trade-off between prediction $I(Z; Y)$ and compression $I(X; Z)$. $I(.)$ represents the mutual information. Then, for any alternative representation $\tilde{Z}$ such that $I(X; \tilde{Z}) \leq I(X; Z^*)$, it holds that:*

$$I(Z^*; Y) \geq I(\tilde{Z}; Y). \tag{2}$$

*That is, under a fixed information constrain $I(X; Z)$, the optimal representation $Z^*$ maximally preserves the predictive information about the target variable $Y$, even when $X$ is composed of multiple modalities.*

*Proof.* Information bottleneck objective (Tishby et al., 1999), presented as $\mathcal{L} = I(Z; Y) - \beta I(Z; X)$, is initially introduced to compress the information with single modality input $X$. Here, $Z$ is a stochastic representation of $X$, and $Y$ is the target. We assume that the multi-modal inputs $X_1, \ldots, X_N$ can be jointly modeled as a single random vector $X = (X_1, \ldots, X_N)$, with a joint distribution $p(x_1, \ldots, x_N, y)$ (Wu & Goodman, 2018). Hence, this assumption allows the mutual information terms $I(Z; Y)$ and $I(Z; X)$ of multi-modal input to be defined in the same way as in the Information Bottleneck objective of single modal input. Given this, the objective of multi-modal information bottleneck is structurally identical to the IB objective defined for single-modal data. Thus, the optimal representation $Z^*$ satisfies the same optimality condition as Eq. 2. Hence, the optimality result from the single modality IB theorem carries over directly to the multi-modal cases. □

**Remark 1** (Dominant and Non-Dominant Modalities in Multimodal Learning). *In multimodal representation learning, modalities contribute differently to predicting the target variable $Y$. A dominant modality $X_d$ is identified when it possesses a higher mutual information with $Y$, denoted as $I(X_d; Y)$, i.e., $X_d = \arg \max_i I(X_i; Y)$. Otherwise, modalities with lower mutual information are considered supportive since it provides less useful information to the prediction.*

*In the brain tumor segmentation, for instance, MRI sequences such as Flair are often more dominant for whole tumor segmentation, whereas T1ce is crucial for enhancing tumor details. Recent*

*empirical results illustrate that dominant modalities consistently outperform non-dominant modalities across key tumor subregion segmentation tasks (Ding et al., 2021).*

*Existing parallel fusion strategies process modalities independently and fuse them at a specific layer; However, they may rely heavily on dominant modalities. As such, prediction performance degrades sharply when the dominant modality is absent, as the fused representation lacks the most informative source. In contrast, Information Bottleneck (IB)-based sequential fusion strategies compress information from multiple modalities into a shared latent representation. Since it preserves task-relevant representation, the fused feature will not rely on the availability of any single dominant input.*

**Theorem 2** (Generalization Bound with Complexity Measures). *(Shalev-Shwartz & Ben-David, 2014; Xu & Raginsky, 2017) Let $\mathcal{H}$ be a hypothesis class, and $h \in \mathcal{H}$ be a learned hypothesis (possibly based on an intermediate representation $Z$ of the input $X$). Given a training set $S = \{(x_i, y_i)\}_{i=1}^n$ drawn i.i.d. from distribution $\mathcal{D}$ where $n$ is the number of samples, in classical learning theory (Shalev-Shwartz & Ben-David, 2014), the generalization error satisfies,*

$$\epsilon_T(h) \leq \epsilon_S(h) + \mathcal{O}\left(\sqrt{\frac{C(\mathcal{H})}{n}}\right),$$

*where the complexity term $C(\mathcal{H})$ denotes a hypothesis class complexity measure such as VC dimension or Rademacher complexity. In information-theoretic learning frameworks, when $Z$ is a learned representation from function $f(.)$, $C(\mathcal{H}, Z)$ can be bounded by the mutual information between $Z$ and the input $X$, i.e., $I(Z; X)$ (Xu & Raginsky, 2017). The generalization error satisfies,*

$$\epsilon_T(h) \leq \epsilon_S(h) + \mathcal{O}\left(\sqrt{\frac{I(\mathcal{Z}, X)}{n}}\right), Z = f(X).$$

The generalization bound is adapted from classical learning theory (Shalev-Shwartz & Ben-David, 2014) and its information-theoretic extension (Xu & Raginsky, 2017). In the following Proposition, the mutual information-based bound is used for comparison since different kinds of fusion appears at the intermediate representation $Z$.

**Proposition 1** (Tighter Generalization Bound via IB Fusion). *Let $Z_{IB} \sim p_{IB}(z|x)$ be the representation produced by an Information Bottleneck (IB)-based fusion model, and $Z_p \sim p_p(z|x)$ be the representation from a parallel fusion model (e.g., concatenation or multi-head attention without a bottleneck). Define the empirical risk gap $\Delta \triangleq \epsilon_S(h_{IB}) - \epsilon_S(h_p)$, and the mutual information gap $g \triangleq I(Z_p; X) - I(Z_{IB}; X)$. Assume that the IB model compresses task-irrelevant information such that $g > 0$ (i.e., $I(Z_{IB}; X) < I(Z_p; X)$), and that the mutual-information generalization inequality in Theorem 2 holds with the same sample size $n$ and hidden constant $c > 0$: $\epsilon_T(h) \leq \epsilon_S(h) + c\sqrt{\frac{I(Z;X)}{n}}$. Then the difference in generalization bounds satisfies*

$$\epsilon_T(h_{IB}) - \epsilon_T(h_p) \leq \Delta - c\left(\sqrt{\frac{I(Z_p; X)}{n}} - \sqrt{\frac{I(Z_{IB}; X)}{n}}\right).$$

*In particular, the IB model achieves a strictly tighter upper bound on the test error whenever the threshold condition*

$$\Delta < c\left(\sqrt{\frac{I(Z_p; X)}{n}} - \sqrt{\frac{I(Z_{IB}; X)}{n}}\right)$$

*is satisfied. It is typically seen since IB compresses mainly task-irrelevant information. Meanwhile, $\Delta$ can be a small value when $I(Z_p; X) - I(Z_{IB}; X)$ is positive.*

This proposition is proved in Appendix Sec. A.

The above proof holds under the assumption that the modalities are independent. However, our analysis can be extended to the case where the modalities are not independent, i.e., when they follow a joint distribution. If the following assumption hold:

**Assumption 1** (Relaxed Cross-Modal Conditional Information Assumption). *The aggregated cross-modal conditional information preserved by the fusion satisfies*

$$\sum_{i=1}^n I(Z_p; X_{<i} \mid X_i) \geq \sum_{i=1}^n \left(2I(Z_{IB}; X_{\neq i}) - I(Z_{IB}; X_i \mid X_{\neq i})\right), \tag{3}$$

*where $Z_p$ denotes the latent representation obtained by ordered fusion, with $X_{<i}$ representing all modalities before the $i$-th modality, and $Z_{IB}$ denotes the latent representation obtained by random fusion, with $X_{\neq i}$ representing all modalities except the $i$-th one.*

A formal justification of this assumption is provided in Appendix Sec. B. Based on this assumption, we have the new relaxed theory:

**Proposition 2.** *Under the relaxed cross-modal conditional information assumption, the following inequality holds for the aggregated mutual information:*

$$I(Z_p; X_1, \ldots, X_n) \geq I(Z_{IB}; X_1, \ldots, X_n),$$

*where $Z_p$ is the latent representation obtained by parallel fusion, and $Z_{IB}$ is the latent representation obtained by the information bottleneck (IB) fusion.*

This proposition is proved in Appendix Sec. C

Please note that the Proposition 1 and Proposition 2 still holds when some of the modalities are missing. In the following, we will draw remarks based on the different influences on $I(Z; X)$ brought by missing modalities.

**Case 1:** $X_s$ **is missing.** Parallel fusion directly maps $Z_p = f(X_d)$ without controlling irrelevant features, leading to relatively high $I(Z_p; X_d)$. Sequential IB-based fusion compresses $X_s$ but leaves $X_d$ unaltered. When $X_s$ is absent, the learned representation $Z_{IB}$ retains only the clean signal from $X_d$, yielding $I(Z_{IB}; X) < I(Z_p; X)$.

**Case 2:** $X_d$ **is missing.** Parallel fusion maps $Z_p = f(X_s)$ without any regularization. As remaining $X_s$ might be less informative to the prediction, it leads to a high $I(Z_p; X_s)$. In contrast, IB-based fusion explicitly compresses $X_s$ while preserving task-relevant information, even without $X_d$, $Z_{IB}$ encodes minimal yet informative patterns: $I(Z_{IB}; X) < I(Z_p; X)$.

As the result, in both cases, $\epsilon_T(h_{IB}) \leq \epsilon_T(h_p)$ will hold. It proves that the generalization upper bound for the sequential IB-based fusion model remains strictly tighter than those with parallel fusion architectures. In the following, we will discuss how IB-based fusion demonstrates a tighter Lipschitz bound that promotes robustness across different missing modality scenarios.

**Assumption 2** (1-Lipschitz Continuity of Modular Components). *Fusion-based missing modality learning framework consists of following modules: (i) an encoder $f_i : X_i \to Z_i$ for each modality, (ii) Let $\phi : Z_1 \times \cdots \times Z_M \to Z_{fused}$ be the multimodal fusion function that aggregates all encoded modality features $\{Z_i\}_{i=1}^M$ into a joint representation $Z_{fused}$, and (iii) a decoder $g : Z^{fused} \to Y$ for prediction. We assume all such components: encoder ($L_i$), fusion ($L_h$ for parallel fusion and $L_{\phi_i}$ for sequential fusion), and decoder ($L_g$) are 1-Lipschitz continuous and all larger than zero.*

**Remark 2.** *While achieving strict 1-Lipschitz continuity can be challenging for softmax-based self-attention, all baselines in our study apply LayerNorm directly after the fusion module. This stabilizes gradients and prevents their uncontrolled growth. Since such normalization can be easily incorporated into similar architectures, the required theoretical conditions remain practically attainable.*

A formal justification of this assumption is provided in Appendix Sec. D. Based on Assumption 2, we provide the Lipschitz bound of parallel and sequential fusions in Appendix Sec. E. In the following Proposition, we would like to compare bounds of different fusions.

**Proposition 3** (Bound of Different Fusion Methods). *Based on Assumption 2, the Lipschitz constant of the parallel and sequential fusion model satisfies: $L_F^{parallel} \leq L_g \cdot L_h \cdot \sqrt{\sum_{i=1}^M L_i^2}$, $\quad L_F^{sequential} \leq L_g \cdot \prod_{i=1}^M L_{\phi_i} \cdot L_i$. If each encoder and fusion module are normalized such that $L_i \leq 1$ and $L_{\phi_i} \leq 1$, then: $\prod_{i=1}^M L_{\phi_i} \cdot L_i \leq \min_i L_i \leq \sqrt{\sum_{i=1}^M L_i^2}$, which implies that, under these conditions, the Lipschitz bound for the sequential fusion model is tighter than that of the parallel fusion model.*

Proposition 3 is proved in the Appendix Sec. F. It establishes that the sequential IB-based fusion model satisfies a tighter Lipschitz continuity constraint, which encourages a smoother decision boundary and contributes to greater robustness.

To empirically validate the robustness across fusion strategies, we visualize and compare the loss landscapes across various missing modality scenarios of: parallel fusion (including concatenation-based fusion and attention-based fusion), and the proposed sequential IB-based fusion. As illustrated

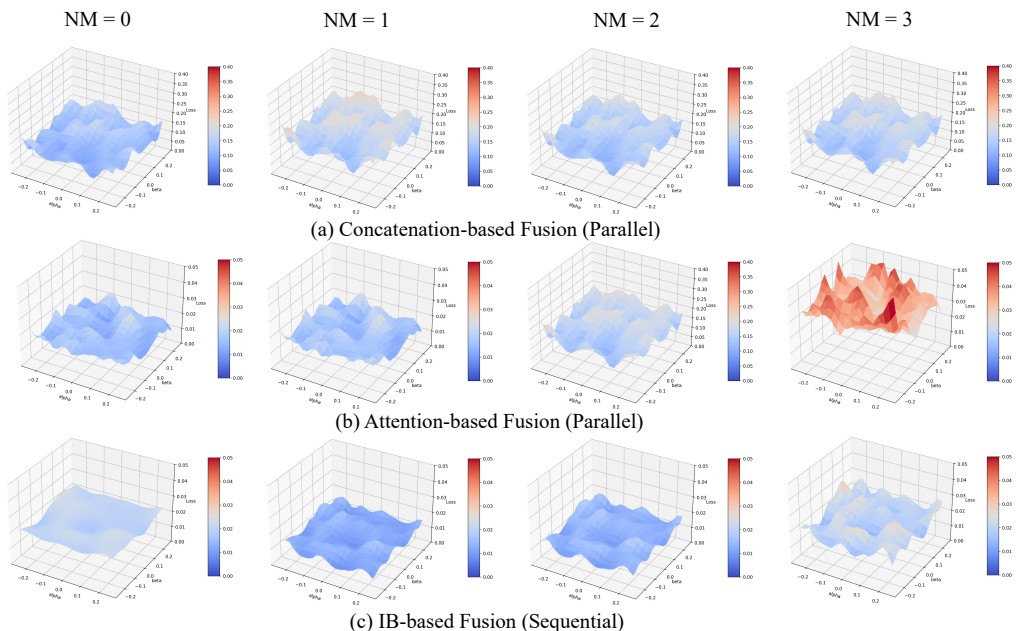

Figure 2: Visualization of the loss landscapes across different missing modality scenarios (NM denotes the number of missing modalities) for Concatenation-based Fusion, Attention-based fusion, and IB-based fusion. Warm colors represent smooth plains, while cold colors depict sharp terrain in the landscape. The bar range for Concatenation-based Fusion is $[0, 0.4]$, while those for Attention-based and IB-based ones are $[0, 0.05]$.

in Figure 2, the sequential IB-based fusion model consistently produces smoother and more stable loss surfaces compared to the baselines. Notably, as the NM increases, the loss landscapes of parallel fusion models become increasingly irregular and exhibit sharper curvature. In contrast, the sequential IB-based fusion model maintains a flatter and more structured loss geometry, even with high NM values. These observations match theoretical findings in Proposition 3, demonstrating that the sequential IB-based fusion model achieves a tighter Lipschitz bound and thus improved smoothness.

## 4 METHODOLOGY

In multi-modal learning, information encoded in each modality can be decomposed into two irrelevant components that would be orthogonal between one and the other in high-dimensional feature space: a modality-specific component and a modality-common one (Zhao et al., 2023). The modality-specific component one captures uniqueness in each modality. In contrast, the modality-common factor encapsulates information that is consistent and shared across multiple modalities. They are afterwards aggregated together to reduce possible information redundancy that might exist in the original modality feature representation (Tsai et al., 2018).

The proposed Sequential Multi-modal Segmentation Network (SMSN) consists of a two-stage information bottleneck fusion module, a specific feature extraction module, and a loss of orthogonality, to achieve the objective that separating modality-specific and modality-common representations of each modality. In particular, based on our discussions in Sec. 3, such sequential information bottleneck fusion strategy processes appealing robustness and generalization properties. They are afterwards aggregated and sent to the decoder, to perform segmentation. [1]

**Modality Reordering Strategy.** We adopt $N$ dedicated modality encoders to extract modality features, corresponding to $N$ modalities. Following the Information Bottleneck objective as Eq. 1, these features are sequentially fused into a shared representation (see *Two-Stage Information Bot-*

---

[1]Please note that the architectures of both the encoder and the decoder follow configurations of the mm-Former (Zhang et al., 2022) and the M2FTrans (Shi et al., 2023).

*tleneck Fusion*), starting from an initial reference modality. However, placing a missing modality (represented as a zero tensor) at the beginning of the sequence may degrade the IB objective. To address it, we propose a modality reordering strategy: one available modality is randomly selected as the initial reference, while the remaining $N - 1$ modalities, regardless of availability, are randomly re-ordered and fused sequentially.

**Two-Stage Information Bottleneck Fusion.**
Inspired by the ITHP (Xiao et al., 2024), we design a two-stage Information Bottleneck Fusion Module to extract modality-common representations. Specifically, given four input modalities, denoted as $x = \{x_i\}_{i=1}^4{}^2$, we plan the fusion process into two stages: the first stage fuses modalities $x_1$ and $x_2$, and the second stage fuses $x_3$ and $x_4$ based on the output of the first stage. Specifically, as illustrated in Fig. 3, we can obtain two bottleneck representations, $z_1$ and $z_2$, with each of them containing compressed latent representations from both stages.

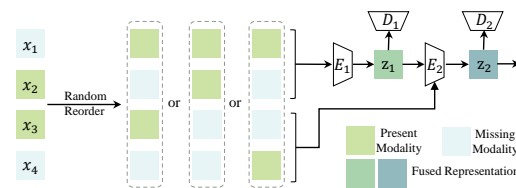

Figure 3: $x = \{x_i\}_{i=1}^4$ are the input modality representations. They are re-ordered and fused by using a two-stage information bottleneck.

The resulted fusion objective $\mathcal{F}$ is formulated with the Information Bottleneck (IB) objective as:

$$\mathcal{F} = \underbrace{I([x_1, x_2]; z_1) - \beta\, I(z_1; y_0)}_{\text{stage I}} + \underbrace{(I(z_1, [x_3, x_4]; z_2) - \gamma\, I(z_2; y_1))}_{\text{stage II}}, \tag{4}$$

where $y_0$ and $y_1$ are the task-related targets for each stage. $\beta, \gamma$ are hyperparameters controlling the trade-off between compression and relevance, refer to Eq. 1. Inspired by the variational approximation of the Information Bottleneck objective (Alemi et al., 2017), the mutual information terms $I([x_1, x_2]; z_1)$ and $I(z_1, [x_3, x_4]; z_2)$ are approximated by Kullback–Leibler divergences ($D_{\mathrm{KL}}$) (Kullback & Leibler, 1951) with respect to a tractable prior. The loss $\mathcal{L}_e$ is formulated as:

$$\mathcal{L}_e = \mathbb{E}_{p([x_1, x_2])}\left[D_{\mathrm{KL}}(p(z_1|[x_1, x_2]) \,\|\, r(z_1))\right] + \mathbb{E}_{p(z_1, [x_3, x_4])}\left[D_{\mathrm{KL}}(p(z_2|z_1, [x_3, x_4]) \,\|\, r(z_1))\right] \tag{5}$$

where $r(.)$ is typically a standard normal prior.

Additionally, two individual decoders will be employed by collecting $z_1$ and $z_2$ respectively to reconstruct input modalities from the compressed latent representations to preserve task-relevant information. To adapt to the missing modality task, we introduce a modality-aware reconstruction loss by multiplying the decoder output with a binary availability mask $M_i \in \{0, 1\}$, where $M_i = 1$ indicates that modality $x_i$ is present. Thus, the reconstruction is:

$$\mathcal{L}_r = \beta \mathbb{E}_{z_0}\left[\sum_{i=1}^{2} M_i \cdot \log q_{\psi_0}(x_i \mid z_0)\right] + \gamma \mathbb{E}_{z_1}\left[\sum_{i=3}^{4} M_i \cdot \log q_{\psi_1}(x_i \mid z_1)\right], \tag{6}$$

where $q_\psi(x_i \mid z)$ denotes the decoder that reconstructs modality $x_i$ from the latent representation $z$. This reconstruction strategy ensures that the network focuses only on reconstructing the modalities that are available, improving robustness under missing modality conditions.

**Specific Feature Extraction.** We employ transformer blocks to disentangle modality-specific information by collecting the concatenation of modality features extracted from $N$ modality encoders $x = \{x_i\}_{i=1}^4$ and the fused representation $z_2$. After this, the output will be split to retrieve the updated modality features corresponding to each of the four input modalities as $N$ modality-specific components $x_{s_i}$. Here, we expect $X_{s_i}$ will contain information that is not properly included in $B_{II}$. To achieve it, we introduce the orthogonality loss: $\mathcal{L}_o = \sum_{i=1}^{M}\left(z_2 \cdot x_s^i\right)^2$. Then, we can obtain features of each involved modality by $x_i^{'} = x_{s_i} + z_2$.

**Final Loss.** Following (Zhang et al., 2022; Shi et al., 2023), we encapsulate all the segmentation losses into $\mathcal{L}_s$, in which $\mathcal{L}_e$, $\mathcal{L}_o$, and $\mathcal{L}_r$ are also jointly optimized.

---

[2]As discussed in Sec. 5, in datasets that we used in this paper, four modalities are involved: T1, T1ce, T2, and Flair.

Table 2: Missing modality segmentation results of different fusion-based models on BRATS18 and BRATS20. For each scenario, from left to right, the four circles represent T2, T1ce, T1, and Flair, respectively. ● represents a modality is present while ○ represents a modality is missing. The best results are highlighted in **bold**.

| | Modality | ●○○○ | ○●○○ | ○○●○ | ○○○● | ●●○○ | ○●●○ | ○○●● | ○●○● | ●○●○ | ●○○● | ○●●● | ●○●● | ●●○● | ●●●○ | ●●●● | Avg. |
|---|---|---|---|---|---|---|---|---|---|---|---|---|---|---|---|---|---|
| | | | | | | | | BRATS18 | | | | | | | | | |
| WT | mmFormer | 84.66 | 73.05 | 73.40 | 86.61 | 85.24 | 76.84 | **88.54** | 84.93 | 89.18 | 88.58 | 88.54 | 89.46 | **89.70** | 85.31 | **89.20** | 84.88 |
| | M²FTrans | 84.11 | 73.86 | 76.88 | **88.16** | 85.52 | 79.04 | 88.34 | 85.63 | **89.22** | **88.88** | 88.41 | 89.20 | 88.74 | 85.95 | 88.87 | 85.39 |
| | MMMViT | 77.92 | 70.38 | 70.70 | 79.22 | 78.77 | 73.67 | 80.57 | 79.29 | 81.51 | 80.46 | 80.83 | 82.03 | 81.66 | 79.19 | 81.86 | 78.54 |
| | IMS²Trans | 83.55 | 68.52 | 67.12 | 85.40 | 84.81 | 71.86 | 87.79 | 84.22 | 89.05 | 88.18 | 87.85 | 89.30 | 89.60 | 84.74 | 88.53 | 83.37 |
| | Ours | **85.49** | **76.33** | **76.26** | 84.38 | **86.45** | **79.62** | 87.82 | **86.61** | 88.51 | 88.71 | **89.09** | **89.98** | 86.93 | **89.39** | 88.71 | 85.62 |
| TC | mmFormer | 64.11 | 78.04 | 62.90 | 60.93 | 79.42 | 79.90 | **68.76** | **69.17** | 65.67 | 79.34 | 80.82 | 70.08 | 79.37 | 79.93 | 80.33 | 73.25 |
| | MMMViT | 59.60 | 75.62 | 63.48 | 60.49 | 77.61 | 77.89 | 66.58 | 65.11 | 64.51 | 77.48 | 78.43 | 67.98 | 78.39 | 78.69 | 78.95 | 71.39 |
| | IMS²Trans | 58.23 | 72.48 | 49.90 | 59.18 | 75.26 | 73.83 | 64.89 | 62.96 | 66.16 | 76.70 | 76.70 | 66.27 | 77.26 | 75.02 | 76.60 | 68.76 |
| | M²FTrans | 62.18 | 78.03 | **65.91** | **62.02** | 79.04 | 80.51 | 67.39 | 67.83 | 65.98 | 78.13 | 79.11 | 68.04 | 78.62 | 79.66 | 79.28 | 72.78 |
| | Ours | **68.27** | **79.72** | 62.34 | 61.85 | **83.57** | **80.75** | 67.70 | 68.55 | **70.78** | **82.35** | **83.17** | **70.77** | **82.81** | **82.96** | **82.35** | 75.20 |
| ET | mmFormer | 36.93 | 70.93 | 30.72 | 30.70 | 70.76 | 69.54 | 35.40 | 37.51 | 38.50 | 71.13 | 72.49 | 38.70 | 70.91 | 69.92 | 70.94 | 54.34 |
| | M²FTrans | 39.51 | 67.87 | 32.40 | 34.03 | 69.92 | 71.24 | 39.41 | 39.07 | 41.15 | 68.62 | 68.89 | 42.96 | 69.70 | 70.03 | 69.44 | 54.95 |
| | MMMViT | 37.25 | 65.46 | 35.07 | 41.82 | 67.57 | 66.38 | 43.44 | 39.19 | 42.68 | 70.97 | 68.85 | 41.59 | 69.93 | 67.87 | 70.08 | 55.21 |
| | IMS²Trans | 40.70 | 67.28 | 24.75 | 31.48 | 70.75 | 66.37 | 38.92 | 42.20 | 42.27 | 70.82 | 68.51 | 41.69 | 70.86 | 70.85 | 71.02 | 54.56 |
| | Ours | **44.51** | **78.87** | **36.55** | **39.31** | **79.70** | **79.37** | **44.42** | **46.25** | **46.09** | **78.58** | **80.30** | **45.71** | **78.91** | **78.70** | **78.58** | 62.39 |
| | | | | | | | | BRATS20 | | | | | | | | | |
| WT | mmFormer | 83.88 | 75.80 | 75.11 | 86.84 | 86.57 | 79.71 | 88.71 | 86.10 | 89.22 | 88.82 | 89.40 | 89.50 | 89.70 | 86.90 | 89.85 | 85.74 |
| | MMMViT | 80.67 | 72.10 | 72.10 | 82.60 | 81.82 | 75.18 | 84.08 | 82.25 | 84.89 | 83.06 | 83.92 | 85.16 | 84.88 | 82.09 | 84.87 | 81.31 |
| | IMS²Trans | 84.35 | 72.91 | 70.83 | 86.50 | 86.51 | 75.58 | 88.94 | 86.64 | 89.43 | 88.16 | 88.96 | 89.99 | 89.93 | 87.27 | 90.27 | 85.08 |
| | M²FTrans | 84.75 | 76.07 | 75.21 | **87.12** | 86.82 | 79.46 | 87.90 | 85.55 | 88.44 | 88.17 | 88.22 | 88.49 | 88.99 | 87.28 | 88.97 | 85.43 |
| | Ours | **87.03** | **78.12** | **77.44** | 87.11 | **88.17** | **82.03** | **89.59** | **88.14** | **90.20** | **89.69** | **89.86** | **90.07** | **90.63** | **88.45** | **90.50** | 87.14 |
| TC | mmFormer | 69.50 | 82.83 | 63.54 | 68.02 | 84.27 | 84.21 | 72.33 | **72.11** | 72.87 | 83.82 | 84.90 | 74.22 | 84.42 | 84.91 | 84.85 | 77.79 |
| | MMMViT | 67.12 | 79.98 | 60.19 | 68.18 | 81.93 | 80.57 | 70.73 | 68.11 | 71.39 | 82.45 | 82.03 | 71.50 | 82.98 | 81.89 | 82.75 | 75.45 |
| | IMS²Trans | 66.39 | 80.14 | 56.09 | 62.15 | 84.94 | 81.06 | 67.80 | 67.20 | 69.58 | 83.69 | 83.30 | 69.88 | 84.57 | 84.67 | 84.28 | 75.05 |
| | M²FTrans | 68.62 | 82.16 | 62.77 | **68.94** | 84.55 | 83.32 | 72.40 | 71.30 | 73.24 | 84.50 | 85.07 | 73.96 | 85.19 | 85.19 | 85.39 | 77.77 |
| | Ours | **70.65** | **83.46** | **65.15** | 68.17 | **85.20** | **84.45** | **73.75** | 71.49 | **73.84** | **85.96** | **85.67** | **75.44** | **86.27** | **85.62** | **86.90** | 78.80 |
| ET | mmFormer | 42.56 | 76.95 | 31.80 | 39.61 | 76.09 | 76.60 | 41.48 | 43.94 | 44.98 | 76.27 | 76.72 | 46.81 | 75.21 | 76.62 | 75.67 | 60.09 |
| | MMMViT | **47.52** | 69.99 | **43.25** | **47.60** | 73.69 | 73.38 | **49.03** | **50.05** | **53.27** | 72.90 | 74.98 | **53.68** | 73.57 | 75.34 | 73.95 | 62.15 |
| | IMS²Trans | 44.25 | 74.90 | 30.77 | 37.54 | 77.50 | 76.73 | 41.95 | 44.12 | 50.50 | 78.85 | 79.46 | 49.97 | 79.50 | 78.40 | 79.85 | 61.62 |
| | M²FTrans | 44.53 | 76.33 | 38.18 | 39.15 | 78.06 | 77.33 | 42.85 | 45.96 | 46.15 | 79.00 | 80.63 | 47.89 | 77.87 | 79.66 | 79.09 | 62.17 |
| | Ours | 45.88 | **78.98** | 38.67 | 40.46 | **79.50** | **76.86** | 43.07 | 46.32 | 49.21 | **80.54** | **80.09** | 47.63 | **79.88** | **79.75** | **79.74** | 63.06 |

# 5 EXPERIMENTS

## 5.1 ROBUSTNESS ON BRATS MRI

**Experimental Settings.** To validate the robustness of the proposed SMSN, we perform comparisons on gliomas segmentation tasks on the BRATS18 and BRATS20 (Menze et al., 2014) for the Multimodal Brain Tumor Segmentation Challenge without any pre-training performed. Specifically, four modalities (T1, T1ce, T2, and Flair) are involved in both datasets. We employ the Dice similarity coefficient as the evaluation metric. For fair comparison, we follow data splits of both sets from M²FTrans (Shi et al., 2023). We also reproduced the results of each fusion-based baseline with the respective released codes.

**Segmentation Results.** We present the comparison results, including the proposed SMSN and other fusion-based baselines in Table 2, demonstrating robustness brought by the SMSN in handling missing modality scenarios. Specifically, SMSN consistently outperforms other baselines in both datasets by evaluating the averaged segmentation performance. Meanwhile, in each missing modality scenario, the proposed SMSN still demonstrates superiority. Importantly, it particularly excels in more challenging scenarios when two or three modalities are not present. It is worth mentioning that although MMViT predicts better ET than SMSN in certain cases on the BRATS20 dataset, the average performance does not demonstrate consistency. We think it is a dataset-sensitive approach, as in the BRATS18 dataset, we can only observe suboptimal performance with the MMViT model.

Except for fusion-based approaches, the proposed SMSN has been proven to outperform non-fusion methods such as M³AE (Liu et al., 2023) and ShaSpec (Wang et al., 2023). Relevant results are presented in the Appendix Table 6.

**Grad-Cam Visualization.** Figure 4 presents Grad-CAM visualizations on the fused feature to illustrate attention focus to predict tuomors on parts of the source image. Specifically, we present results of the proposed SMSN and M²FTrans that performs relatively better than other baselines in Table 2. From these heatmaps, we can observe that the proposed SMSN achieves a more focused activation pattern, which also well aligns with the ground truth depicted by orange and red contours.

This finding supports our discussion in Remark 1 that with the proposed SMSN, even though some certain modality is missing, the fusion is still able to focus on the task-relevant representation and the prediction task.

## 5.2 GENERALIZATION FROM GLIOMAS TO METASTASES

**Experiment Settings.** Except for gliomas, metastases are another kind of malignant brain tumor. As seen in Sec. G.6, they differ greatly in morphology. Gliomas appear as infiltrative lesions with indistinct margins and heterogeneous enhancement, whereas metastases are typically well-circumscribed, round or ovoid with sharp borders and marked surrounding edema. It is challenging to generalize a prediction model trained only with gliomas to metastases predictions. Here, we employ trained SMSN and other relevant baselines only with BRATS20 to the Brain Metastases (BM) dataset (Ramakrishnan et al., 2024) with identical modalities: T1, T1ce, T2, and Flair.

**Segmentation Results** As presented in Table 3, the proposed SMSN consistently outperforms other fusion-based methods, which proved appealing generalization performance. Furthermore, compared to the second-best $M^2$FTrans model, SMSN exhibits lower standard deviations, indicating enhanced robustness in varying input conditions. The detailed results of Table 3 with each individual scenario is presented in the Appendix Table 7.

## 5.3 ABLATION STUDIES

Table 4 presents ablation studies of the proposed SMSN by evaluating prediction performance upon removing each constructing individual module/loss discussed in Sec. 4. Notably, the orthogonality loss will is in line with the specific feature extraction module, it will also be removed when that module is absent. These obtained results demonstrate that each constructing module/loss is essential to the final prediction result.

It is important to be noticed that IB theoretically can decompose common information from a mixed feature. However, in real practice, some modality-specific information will still be preserved in the IB module. To promote modality decomposition, the specific feature extraction module and the orthogonal loss is additionally applied. Based on acquired results from the ablation study, canceling orthogonal loss degrades segmentation performance, which confirms our aforementioned analysis.

Table 3: Segmentation results under missing-modality conditions on the meta dataset, with models trained on BRATS. NM denotes the number of missing modalities, and FM indicates all modalities are available. Reported values correspond to the mean and standard deviation of Dice scores for each missing-modality scenario.

| Modality | | NM=3 | | NM=2 | | NM=1 | | FM | Avg. |
|---|---|---|---|---|---|---|---|---|---|
| | | Mean | Std. | Mean | Std. | Mean | Std. | Mean | Mean |
| WT | mmFormer | 32.53 | 10.71 | 30.85 | 9.70 | 18.55 | 12.72 | 37.15 | 28.43 |
| | $M^2$FTrans | 49.99 | 5.73 | 55.40 | 3.94 | 58.87 | 3.43 | 59.58 | 55.16 |
| | MMMViT | 46.82 | 3.76 | 50.31 | 2.23 | 51.90 | 0.65 | 52.72 | 49.96 |
| | IMS$^2$Trans | 48.95 | 6.96 | 55.67 | 4.62 | 58.70 | 2.51 | 60.14 | 54.98 |
| | Ours | 51.69 | 4.93 | 57.64 | 4.25 | 60.33 | 2.36 | 61.55 | 57.03 |
| TC | mmFormer | 10.34 | 5.19 | 9.30 | 5.39 | 5.18 | 5.90 | 12.50 | 8.69 |
| | $M^2$FTrans | 28.83 | 8.67 | 38.48 | 10.01 | 45.44 | 8.88 | 51.36 | 38.62 |
| | MMMViT | 27.52 | 10.86 | 36.04 | 11.42 | 43.13 | 9.88 | 48.64 | 36.50 |
| | IMS$^2$Trans | 28.04 | 13.58 | 38.60 | 12.75 | 47.10 | 10.61 | 53.51 | 39.04 |
| | Ours | 35.50 | 12.98 | 45.37 | 13.21 | 53.03 | 10.56 | 59.17 | 45.70 |
| TC | mmFormer | 6.85 | 5.66 | 5.68 | 4.03 | 4.63 | 4.07 | 6.73 | 5.78 |
| | $M^2$FTrans | 21.34 | 11.31 | 31.24 | 12.92 | 39.16 | 11.83 | 46.67 | 31.74 |
| | MMMViT | 21.90 | 12.89 | 29.54 | 14.00 | 37.73 | 12.48 | 43.86 | 30.64 |
| | IMS$^2$Trans | 21.90 | 12.89 | 29.54 | 14.00 | 37.73 | 12.48 | 43.86 | 30.64 |
| | Ours | 26.07 | 14.12 | 36.43 | 14.65 | 44.69 | 13.69 | 51.72 | 36.89 |

Table 4: Ablation results of models trained under different combinations of modules/losses on BRATS20. From left to right, the four circles represent modality reordering strategy, $\mathcal{L}_r$, specific feature extraction module, and $\mathcal{L}_o$, respectively. ● represents a modality is module/loss while ○ represents a module/loss is missing.

| Class | ○●●● | ●○●● | ●●○○ | ●●●○ | ●●●● |
|---|---|---|---|---|---|
| WT | 85.75 | 85.65 | 85.57 | 85.69 | 87.14 |
| TC | 76.38 | 76.82 | 76.19 | 76.61 | 78.80 |
| ET | 63.09 | 62.16 | 62.36 | 63.19 | 63.00 |

Full Grad-Cam    NM = 1 Grad-Cam (Ours)    NM = 1 Grad-Cam M2FTrans

Figure 4: This figure presents a comparative visualization of Grad-CAM heatmaps.

## 5.4 SENSITIVE ANALYSIS

Referring to Eq. 5, $\beta, \gamma$ are hyper-parameters controlling the trade-off between compression and relevance. As shown in Table 5, our experiments indicate that the performance of the information bottleneck compression is indeed sensitive to the choice of the hyper-parameters $\gamma$ and $\beta$. This sensitivity is expected because these parameters directly control the trade-off between compression and task-relevant information preservation: a larger $\beta$ enforces stronger compression, potentially discarding useful features, while a smaller $\beta$ retains more information but may reduce the regularization effect. Similarly, $\gamma$ adjusts the relative weighting of different components in the loss, affecting how strictly the model satisfies the information constraints.

Despite this sensitivity, we observe that across a wide range of $\gamma$ and $\beta$ values, the proposed method consistently outperforms involved baselines, indicating that the information bottleneck framework provides robust gains even when the hyper-parameters are not finely tuned.

Table 5: Sensitive Analysis of hyper-parameters trained on BRATS18 and BRATS20. The two numbers in the first line of each scenario are the values of $\beta$ and $\gamma$, respectively.

| BRATS18 | [0.1, 1] | [0.3, 0.7] | [0.7, 0.3] | [0.5, 0.5] | [1, 0.1] | BRATS20 | [0.1, 1] | [0.3, 0.7] | [0.7, 0.3] | [0.5, 0.5] | [1, 0.1] |
|---|---|---|---|---|---|---|---|---|---|---|---|
| WT | 85.28 | 84.69 | 85.62 | 85.58 | 85.69 | WT | 87.27 | 86.52 | 87.14 | 88.33 | 86.27 |
| TC | 74.19 | 72.72 | 75.20 | 74.14 | 73.55 | TC | 78.00 | 79.22 | 78.80 | 79.94 | 79.72 |
| ET | 56.71 | 61.69 | 62.39 | 57.34 | 57.56 | ET | 62.29 | 63.59 | 63.00 | 61.49 | 60.34 |

## 6 CONCLUSION

We presented the Sequential Multi-modal Segmentation Network (SMSN) to address missing modalities in multi-modal MRI brain tumor segmentation. By leveraging an Information-Bottleneck Fusion Module (IBFM), SMSN sequentially disentangles modality-common features while reconstructing modality-specific information. IBFM provides a Lipschitz-continuous lower bound and a tighter generalization upper bound, improving robustness and cross-domain adaptability. Experiments on BRATS18 and BRATS20 show that SMSN outperforms the fusion-based baselines under various missing-modality settings. Result on transferring the model to a brain metastasis dataset without further fine-tuning confirms its strong generalization capability.

## ACKNOWLEDGE

The work was partially supported by the following: National Natural Science Foundation of China under No. 92370119, and 62376113; Xi'an Jiaotong-Liverpool University Postgraduate Research Scholarship PGRS2206013. Special thanks to Dr. Fan Zhang, from Xi'an Jiaotong-Liverpool University, for the constructive advice on the writing of this paper.

## ETHICS STATEMENT

Our study does not involve human subjects, sensitive data, or personally identifiable information. The datasets we use are publicly available and commonly adopted in the community. We are not aware of potential ethical risks related to privacy, fairness, or misuse.

## REPRODUCIBILITY STATEMENT

We are committed to ensuring the reproducibility of our results. Implementation details, including model architecture and training setup, are provided in Sec 5 of the main paper. We also include the source code and scripts in the supplementary materials to facilitate replication of our results.

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

# A  PROOF OF PROPOSITION 1

*Proof.* From Theorem 2, the MI-based generalization bounds for $h_{IB}$ and $h_p$ are

$$\epsilon_T(h_{IB}) \leq \epsilon_S(h_{IB}) + c\sqrt{\frac{I(Z_{IB}; X)}{n}}, \qquad \epsilon_T(h_p) \leq \epsilon_S(h_p) + c\sqrt{\frac{I(Z_p; X)}{n}}.$$

Subtracting the second inequality from the first gives an upper bound on their difference:

$$\epsilon_T(h_{IB}) - \epsilon_T(h_p) \leq \Delta + c\left( \sqrt{\frac{I(Z_{IB}; X)}{n}} - \sqrt{\frac{I(Z_p; X)}{n}} \right), \quad \Delta = \epsilon_S(h_{IB}) - \epsilon_S(h_p).$$

The second term $c(.)$ in the aforementioned inequality will be negative because the IB objective explicitly penalizes redundant information from the input:

$$\mathcal{L}_{\text{IB}}(h) = \epsilon_S(h) + \beta I(Z; X) - \alpha I(Z; Y), \quad \beta > 0.$$

Thus, $Z_{IB}$ compresses $X$ to preserve only a task-relevant subset, often with even smaller $I(Z_{IB}; X)$ than $I(X; Y)$. In contrast, the parallel fusion representation $Z_p$ does not restrict $I(Z; X)$ and typically retains more task-irrelevant noise, given $I(Z_{IB}; X) < I(Z_p; X)$. Denote $g = I(Z_p; X) - I(Z_{IB}; X) > 0$, then we will obtain

$$c\left( \sqrt{\frac{I(Z_{IB}; X)}{n}} - \sqrt{\frac{I(Z_p; X)}{n}} \right) = -\frac{c\,g}{\sqrt{n}(\sqrt{I(Z_p; X)} + \sqrt{I(Z_{IB}; X)})} < 0.$$

The threshold condition

$$\Delta < c\left( \sqrt{\frac{I(Z_p; X)}{n}} - \sqrt{\frac{I(Z_{IB}; X)}{n}} \right)$$

is typically easy to satisfy: $\Delta$ remains small because the IB-based fusion model primarily discards task-irrelevant information. Empirical risk will not be increased significantly, while $g$ is positive as the parallel fusion model retains additional redundant information. Therefore, the right-hand side of the inequality for $\epsilon_T(h_{IB}) - \epsilon_T(h_p)$ is strictly negative, leading to $\epsilon_T(h_{IB}) < \epsilon_T(h_p)$. Therefore, the IB model achieves a strictly tighter generalization bound. □

# B  JUSTIFICATION OF ASSUMPTION 1

We now explain that this assumption is reasonable in the case of medical multi-modal fusion tasks.

The term $I(Z_p; X_{<i} \mid X_i)$ represents the conditional mutual information between the fusion latent variable $Z_p$ and all other modalities $X_{<i}$, given modality $X_i$. This term measures the amount of information retained by $Z_p$ about the other modalities once $X_i$ is known. The term $I(Z_{IB}; X_i \mid X_{\neq i})$ represents the conditional mutual information in the IB model, capturing the dependence of each modality on the rest of the modalities. We have $I(Z_p; X_{<i} \mid X_i) \geq I(Z_{IB}; X_{\neq i}) - I(Z_{IB}; X_i \mid X_{\neq i})$.

In practice, the modalities in our data are strongly correlated. This means that the parallel fusion model $Z_p$ can preserve more information about the inter-modal relationships compared to the IB fusion model, which is designed to compress information to retain only the most relevant aspects for prediction. As a result, $I(Z_p; X_{<i} \mid X_i)$ tends to be larger than the corresponding terms in the IB model. We may relax the bound to $I(Z_p; X_{<i} \mid X_i) \geq 2I(Z_{IB}; X_{\neq i}) - I(Z_{IB}; X_i \mid X_{\neq i})$.

To empirically validate this assumption, we compute the mutual information for both the parallel fusion and IB fusion models on our dataset. The computed values show that: $\sum_{i=1}^{n} I(Z_p; X_{<i} \mid X_i) = 0.2817$ and $\sum_{i=1}^{n}(-I(Z_{IB}; X_i \mid X_{\neq i}) + I(Z_{IB}; X_{\neq i})) = 0.0709$. These results support the validity of the relaxed bound, demonstrating that the information preserved by the parallel fusion model is indeed larger than or equal to the relaxed bound defined by the IB fusion model.

## C  PROOF OF PROPOSITION 2

*Proof.* We start by expanding the mutual information for both the parallel fusion model $Z_p$ and the information bottleneck (IB) fusion model $Z_{IB}$. For the parallel fusion model, we have the following expansion for the total mutual information:

$$I(Z_p; X_1, \ldots, X_n) = \sum_{i=1}^{n} \left( I(Z_p; X_i) + I(Z_p; X_{<i} \mid X_i) - I(Z_p; X_{<i}) \right).$$

For the information bottleneck model, we expand the total mutual information as:

$$I(Z_{IB}; X_1, \ldots, X_n) = \sum_{i=1}^{n} \left( I(Z_{IB}; X_i) - I(Z_{IB}; X_i \mid X_{\neq i}) + I(Z_{IB}; X_{\neq i}) \right).$$

We have two more Inequalities: Information of a single modality preserved by parallel fusion larger than that preserved by IB fusion: $\sum_{i=1}^{n} I(Z_p; X_i) \geq \sum_{i=1}^{n} I(Z_{IB}; X_i)$.

Negative value of the information of the preceding modal set in parallel fusion smaller than that in IB fusion: $-\sum_{i=1}^{n} I(Z_p; X_{<i}) \leq -\sum_{i=1}^{n} I(Z_{IB}; X_{\neq i})$.

Therefore, we can rewrite the total mutual information for the parallel fusion model $Z_p$ as follows:

$$I(Z_p; X_1, \ldots, X_n) = \sum_{i=1}^{n} (I(Z_p; X_i) + I(Z_p; X_{<i} \mid X_i) - I(Z_p; X_{<i}))$$

$$\geq \sum_{i=1}^{n} ( \underbrace{I(Z_{IB}; X_i)}_{I(Z_p; X_i) \geq I(Z_{IB}; X_i)} + \underbrace{[2I(Z_{IB}; X_{\neq i}) - I(Z_{IB}; X_i \mid X_{\neq i})]}_{I(Z_p; X_{<i} \mid X_i) \geq 2I(Z_{IB}; X_{\neq i}) - I(Z_{IB}; X_i \mid X_{\neq i})}$$

$$+ \underbrace{(-I(Z_{IB}; X_{\neq i})))}_{-I(Z_p; X_{<i}) \leq -I(Z_{IB}; X_{\neq i})}$$

$$= \sum_{i=1}^{n} (I(Z_{IB}; X_i) + 2I(Z_{IB}; X_{\neq i}) - I(Z_{IB}; X_i \mid X_{\neq i}) - I(Z_{IB}; X_{\neq i}))$$

$$= \sum_{i=1}^{n} (I(Z_{IB}; X_i) - I(Z_{IB}; X_i \mid X_{\neq i}) + I(Z_{IB}; X_{\neq i}))$$

$$= I(Z_{IB}; X_1, X_2, \ldots, X_n)$$

$\square$

## D  JUSTIFICATION OF ASSUMPTION 2

We decompose the fusion-based missing-modality model into three components: encoders, a fusion function, and a decoder, and show that each satisfies Lipschitz continuity.

For the $i$-th modality, let the input space be $\mathcal{X}_i$ and the encoder $f_i : \mathcal{X}_i \to \mathcal{Z}_i$. Assume that for any $x_i, x_i' \in \mathcal{X}_i$, the encoder is Lipschitz continuous, i.e.,

$$\|f_i(x_i) - f_i(x_i')\| \leq L_i \|x_i - x_i'\|,$$

where $\| \cdot \|$ denotes the norm in the respective space. The fusion function at each step, denoted by $\phi_i : \mathcal{Z}_{i-1} \times \mathcal{Z}_i \to \mathcal{Z}_i^{\text{fused}}$, is also Lipschitz continuous, i.e.,

$$\|\phi_i(z_{i-1}, z_i) - \phi_i(z_{i-1}', z_i')\| \leq L_{\phi_i} \left( \|z_{i-1} - z_{i-1}'\| + \|z_i - z_i'\| \right),$$

and the final decoder $g : \mathcal{Z}_M \to \mathcal{Y}$ is Lipschitz continuous with constant $L_g$.

Since the composition of Lipschitz functions is Lipschitz continuous with constant given by the product of the individual constants, the full function

$$G(x) = g \circ \phi_M \circ \cdots \circ \phi_1 \circ (f_1, \ldots, f_M)(x)$$

is Lipschitz continuous, where $\circ$ represents the function composition. In practice, spectral normalization ensures that each module can be implemented as 1-Lipschitz. Therefore, without loss of generality, the entire missing-modality model can be regarded as 1-Lipschitz, which justifies Assumption 2.

# E    LIPSCHITZ BOUND OF FUSION MODELS

**Proposition 4** (Lipschitz constants of IB fusion). *Suppose that there are $M$ modalities to be fused. Let $F(x_1, \ldots, x_M) := g \circ \phi_M \circ \cdots \circ \phi_2 \circ \phi_1 (f_1(x_1), \ldots, f_M(x_M))$ denote the multi-modal prediction function based on step-wise IB fusion. $\circ$ represents the function composition. Under Assumptions 2, the function $F$ is Lipschitz continuous with Lipschitz constant bounded above by induction on the number of fusion steps: $L_F \le L_g \cdot \prod_{i=1}^{M} L_{\phi_i} \cdot L_i$.*

*Proof.* Let $\tilde{z}_1 = f_1(x_1)$ be the output of the first encoder, which is $L_1$-Lipschitz by Assumption 2.

For the first fusion step, define $\tilde{z}_2 = \phi_1(\tilde{z}_1, f_2(x_2))$. By Assumption 2, $\phi_1$ is $L_{\phi_1}$-Lipschitz in each argument. Since $f_2$ is $L_2$-Lipschitz, the composition satisfies $\tilde{z}_2$ is $L_{\phi_1} \cdot L_1 \cdot L_2$-Lipschitz.

Suppose after $i$ fusion steps, the representation $\tilde{z}_{i+1}$ is Lipschitz with constant

$$L_{i+1} = \left( \prod_{j=1}^{i} L_{\phi_j} \cdot L_j \right) \cdot L_{i+1}.$$

Then for the $(i+1)$-th step,

$$\tilde{z}_{i+2} = \phi_{i+1}(\tilde{z}_{i+1}, f_{i+2}(x_{i+2})).$$

Applying the Lipschitz property of $\phi_{i+1}$ in each argument and the inductive hypothesis, we obtain

$$L_{i+2} \le L_{\phi_{i+1}} \cdot L_{i+1} \cdot L_{i+2}.$$

By induction, after all the $M$ modalities are step-wisely fused, the obtain representation $\tilde{z}_{M+1}$ satisfies the Lipschitz constant

$$\prod_{i=1}^{M} L_{\phi_i} \cdot L_i.$$

Finally, applying the decoder $g$ with Lipschitz constant $L_g$, we can obtain:

$$F(x_1, \ldots, x_M) = g(\tilde{z}_{M+1}).$$

Thus, the upper bound of Lipschitz continuous function $L_F$ with constant will be no greater than $L_g \cdot \prod_{i=1}^{M} L_{\phi_i} \cdot L_i$, which is represented by:

$$L_F \le L_g \cdot \prod_{i=1}^{M} L_{\phi_i} \cdot L_i.$$

This completes the proof. $\qquad\square$

**Proposition 5** (Lipschitz constant of concatenation-based fusion). *Let $F_{\text{concat}}(x_1, \ldots, x_M) := g \circ h_{\text{concat}}(f_1(x_1), f_2(x_2), \ldots, f_M(x_M))$ denote the overall multi-modal prediction function based on concatenation fusion, where $h_{\text{concat}} : \mathcal{Z}_1 \times \cdots \times \mathcal{Z}_M \to \mathcal{Z}_{\text{fused}}$ performs concatenation followed by a mapping (e.g., fully connected layer), and $g : \mathcal{Z}_{\text{fused}} \to \mathcal{Y}$ is the final decoder. Assume each encoder $f_i : \mathcal{X}_i \to \mathcal{Z}_i$ is Lipschitz continuous with constant $L_i$, and $h_{\text{concat}}$. $g$ are Lipschitz continuous with constants $L_{\text{concat}}$ and $L_g$, respectively. Then $F_{\text{concat}}$ is Lipschitz continuous with constant bounded by*

$$L_{F_{\text{concat}}} \le L_g \cdot L_{\text{concat}} \cdot \sqrt{\sum_{i=1}^{M} L_i^2}.$$

*Proof.* Let $z = (f_1(x_1), \ldots, f_M(x_M)) \in \mathcal{Z}_1 \times \cdots \times \mathcal{Z}_M$. Then, by engaging the Euclidean norm,

$$\|z - z'\| = \sqrt{\sum_{i=1}^{M} \|f_i(x_i) - f_i(x_i')\|^2} \le \sqrt{\sum_{i=1}^{M} L_i^2 \|x_i - x_i'\|^2}.$$

By the Lipschitz continuity of $h_{\text{concat}}$ and $g$, we have

$$
\begin{aligned}
\|F_{\text{concat}}(x) - F_{\text{concat}}(x')\| &= \|g(h_{\text{concat}}(z)) - g(h_{\text{concat}}(z'))\| \\
&\leq L_g \|h_{\text{concat}}(z) - h_{\text{concat}}(z')\| \\
&\leq L_g\, L_{\text{concat}}\, \|z - z'\| \\
&\leq L_g\, L_{\text{concat}} \sqrt{\sum_{i=1}^{M} L_i^2 \|x_i - x_i'\|^2}.
\end{aligned}
$$

This proves the stated bound. $\qquad\square$

**Proposition 6** (Lipschitz constant of attention-based fusion). *Let $F_{\text{attn}}(x_1, \ldots, x_M) := g \circ h_{\text{attn}}\big(f_1(x_1), \ldots, f_M(x_M)\big)$ denote the multi-modal prediction function with attention-based fusion, where each encoder $f_i : \mathcal{X}_i \to \mathcal{Z}_i$ is $L_i$-Lipschitz continuous, $h_{\text{attn}}$ is the attention-based fusion, and $g$ is the decoder. Assume $h_{\text{attn}}$ and $g$ are Lipschitz continuous with constants $L_{\text{attn}}$ and $L_g$, respectively. Then $F_{\text{attn}}$ is Lipschitz continuous with constant bounded by*

$$
L_{F_{\text{attn}}} \leq L_g \cdot L_{\text{attn}} \cdot \sqrt{\sum_{i=1}^{M} L_i^2}.
$$

*Proof.* Let $z = (f_1(x_1), \ldots, f_M(x_M)) \in \mathcal{Z}_1 \times \cdots \times \mathcal{Z}_M$. Then

$$
\|z - z'\| = \sqrt{\sum_{i=1}^{M} \|f_i(x_i) - f_i(x_i')\|^2} \leq \sqrt{\sum_{i=1}^{M} L_i^2 \|x_i - x_i'\|^2}.
$$

By the Lipschitz continuity of $h_{\text{attn}}$ and $g$,

$$
\begin{aligned}
\|F_{\text{attn}}(x) - F_{\text{attn}}(x')\| &= \|g(h_{\text{attn}}(z)) - g(h_{\text{attn}}(z'))\| \\
&\leq L_g \|h_{\text{attn}}(z) - h_{\text{attn}}(z')\| \\
&\leq L_g\, L_{\text{attn}}\, \|z - z'\| \\
&\leq L_g\, L_{\text{attn}} \sqrt{\sum_{i=1}^{M} L_i^2 \|x_i - x_i'\|^2}.
\end{aligned}
$$

This proves the stated bound. $\qquad\square$

**Remark 3.** *Both concatenation-based fusion and attention-based fusion are examples of parallel fusion. They share the same Lipschitz constant upper bound:*

$$
L \leq L_g \cdot L_h \cdot \sqrt{\sum_{i=1}^{M} L_i^2},
$$

*where $L_h$ is replaced by $L_{\text{concat}}$ or $L_{\text{attn}}$ depending on respective fusion method.*

## F    PROOF OF PROPOSITION 2

*Proof.* Since all $L_i \leq 1$, their product satisfies: $\prod_{i=1}^{M} L_i \leq \min_i L_i$. By definition of $\ell_2$ norm and $\ell_\infty$ norm, we have: $\min_i L_i \leq \max_i L_i \leq \sqrt{\sum_{i=1}^{M} L_i^2}$. Similarly, since $L_{\phi_i} \leq 1$, we have: $\prod_{i=1}^{M} L_{\phi_i} \leq 1$. Combining them together, we will obtain:

$$
\prod_{i=1}^{M} L_{\phi_i} \cdot L_i \leq \prod_{i=1}^{M} L_i \leq \min_i L_i \leq \sqrt{\sum_{i=1}^{M} L_i^2}.
$$

Multiplying both sides by $L_g$, and noting $L_h \geq 1$ typically for concatenation mappings, gives

$$L_g \cdot \prod_{i=1}^{M} L_{\phi_i} \cdot L_i \leq L_g \cdot \sqrt{\sum_{i=1}^{M} L_i^2} \leq L_g \cdot L_h \cdot \sqrt{\sum_{i=1}^{M} L_i^2}.$$

Therefore, $L_F^{parallel} \leq L_F^{sequential}$. It theoretically evidences that the IB fusion is potentially smoother. □

## G ADDITIONAL EXPERIMENT RESULTS

### G.1 SEGMENTATION RESULTS ON NO-FUSION SOTAS

Table 6 presents the comparisons of segmentation with the proposed SMSN and non-fusion approaches, including Shaspec and m3ae on the BRATS18 and the BRATS20 datasets. SMSN demonstrates consistent improvements across most missing-modality scenarios. In particular, we observe average improvements of about 1 DICE for the WT and TC classes, and more than 5 DICE for the ET class.

Table 6: Missing modality segmentation results of MRI on BRATS18 and BRATS20: Num denotes the number of missing modalities for different settings. NM is the missing number. Each column shows the average dice of different NM. The results of each setting are presented accordingly. The best results are highlighted in red while the second best is highlighted in blue.

| Class | Method | BRATS18 | | | | | BRATS20 | | | | |
|---|---|---|---|---|---|---|---|---|---|---|---|
| | | NM=3 | NM=2 | NM=1 | Full | AVG. | NM=3 | NM=2 | NM=1 | Full | AVG. |
| WT | m3ae | 80.88 | 85.65 | 88.24 | 88.93 | 85.28 | 77.60 | 84.93 | 87.98 | 89.19 | 84.07 |
| | Shaspec | 78.12 | 84.80 | 88.13 | 89.93 | 84.25 | 78.77 | 86.62 | 88.78 | 89.91 | 85.32 |
| | ours | 80.62 | 86.29 | 88.85 | 88.71 | 85.62 | 82.43 | 87.97 | 89.75 | 90.50 | 87.14 |
| ET | m3ae | 68.40 | 75.04 | 79.42 | 81.59 | 74.87 | 70.32 | 77.64 | 81.59 | 84.76 | 77.21 |
| | shaspec | 67.77 | 75.15 | 79.33 | 81.79 | 74.74 | 68.85 | 74.89 | 80.47 | 84.87 | 75.43 |
| | ours | 68.05 | 75.62 | 79.93 | 82.35 | 75.20 | 71.86 | 79.12 | 83.25 | 86.90 | 78.80 |
| TC | mmformer | 42.32 | 53.81 | 63.01 | 70.94 | 54.34 | 47.54 | 56.82 | 64.42 | 71.40 | 57.34 |
| | shaspec | 45.34 | 53.11 | 60.97 | 67.64 | 54.10 | 45.98 | 55.53 | 63.97 | 71.43 | 56.29 |
| | ours | 49.81 | 62.40 | 70.91 | 78.58 | 62.39 | 51.00 | 62.58 | 71.45 | 79.74 | 63.00 |

### G.2 DETAILED GENERALIZATION RESULTS

Missing modality segmentation results of different fusion-based models pretrained on BRATS20 and predicted without any finetuning on the Brain Metastases (BM) dataset is illustrated in Table 7. Our proposed approach consistently outperforms other fusion-based methods across all missing modality configurations, demonstrating its superior generalization performance.

Table 7: Missing modality segmentation results of different fusion-based models pretrained on BRATS20 and predicted without any finetuning on Brain Metastases (BM) dataset. The best results are highlighted in red while the second best is highlighted in blue. ∼ means without. From top to bottom, it illustrates the results of WT, TC, and ET respectively.

| Method | T2 | T1C | T1 | F | T2 T1C | T1 T1C | F T1 | T1 T2 | F T2 | F T1C | ∼T2 | ∼T1C | ∼T1 | ∼F | - | AVG. |
|---|---|---|---|---|---|---|---|---|---|---|---|---|---|---|---|---|
| mmFormer | 32.17 | 36.20 | 18.12 | 43.61 | 27.38 | 15.29 | 32.60 | 44.86 | 34.87 | 30.07 | 24.03 | 4.70 | 33.43 | 12.04 | 37.15 | 28.43 |
| M2FTrans | 51.79 | 45.84 | 45.04 | 57.30 | 53.82 | 48.99 | 58.04 | 53.79 | 59.16 | 58.59 | 58.68 | 62.90 | 59.35 | 54.53 | 59.58 | 55.16 |
| MMVIT | 49.14 | 45.20 | 42.34 | 50.60 | 50.40 | 45.89 | 51.91 | 50.85 | 51.34 | 51.49 | 51.95 | 52.48 | 52.18 | 50.98 | 52.72 | 49.96 |
| IMSTrans | 50.71 | 46.62 | 40.97 | 57.51 | 53.91 | 47.62 | 59.04 | 54.83 | 59.03 | 59.59 | 60.11 | 59.86 | 59.89 | 54.93 | 60.14 | 54.98 |
| Ours | 54.22 | 48.87 | 46.42 | 57.24 | 56.33 | 50.65 | 60.46 | 56.29 | 59.23 | 62.89 | 62.86 | 60.72 | 60.60 | 57.15 | 61.55 | 57.03 |
| mmFormer | 9.46 | 14.35 | 3.35 | 14.21 | 6.60 | 2.63 | 9.01 | 18.84 | 8.22 | 10.51 | 6.78 | 0.07 | 12.83 | 1.04 | 12.50 | 8.69 |
| M2FTrans | 21.81 | 41.09 | 23.73 | 28.67 | 46.52 | 45.60 | 31.57 | 26.17 | 31.06 | 49.95 | 52.18 | 32.49 | 49.96 | 47.13 | 51.36 | 38.62 |
| MMVIT | 22.78 | 43.75 | 20.80 | 22.75 | 46.18 | 43.63 | 27.07 | 23.80 | 26.49 | 49.09 | 49.74 | 28.41 | 47.31 | 47.04 | 48.64 | 36.50 |
| IMSTrans | 23.90 | 47.77 | 16.71 | 23.76 | 50.36 | 47.10 | 24.44 | 27.34 | 29.73 | 52.63 | 53.28 | 31.21 | 52.33 | 51.57 | 53.51 | 39.04 |
| Ours | 28.29 | 54.95 | 28.96 | 29.78 | 56.07 | 55.10 | 34.05 | 32.58 | 33.71 | 60.72 | 60.36 | 37.35 | 57.71 | 56.68 | 59.17 | 45.70 |
| mmFormer | 8.07 | 14.33 | 3.10 | 1.90 | 6.01 | 2.87 | 3.10 | 10.82 | 1.16 | 10.12 | 8.08 | 0.38 | 8.15 | 1.92 | 6.73 | 5.78 |
| M2FTrans | 15.25 | 38.28 | 15.34 | 16.49 | 41.34 | 40.76 | 20.95 | 17.92 | 20.00 | 46.49 | 47.38 | 21.82 | 45.95 | 41.50 | 46.67 | 31.74 |
| MMVIT | 15.96 | 41.19 | 14.35 | 16.08 | 42.14 | 39.63 | 17.34 | 16.29 | 16.94 | 44.92 | 45.22 | 19.08 | 44.07 | 42.56 | 43.86 | 30.64 |
| IMSTrans | 16.04 | 40.74 | 10.86 | 13.17 | 43.56 | 42.26 | 16.90 | 17.89 | 19.27 | 43.02 | 43.62 | 20.78 | 44.01 | 43.57 | 43.60 | 30.62 |
| Ours | 20.26 | 47.17 | 19.30 | 17.53 | 49.07 | 47.05 | 22.84 | 24.13 | 22.55 | 52.92 | 53.22 | 24.25 | 51.27 | 50.01 | 51.72 | 36.89 |

## G.3 SENSITIVE ANALYSIS OF DIFFERENT ORDER

As shown in Table 8, both a fixed ordering and a purely random ordering without ensuring that the first modality corresponds to a present modality lead to degraded segmentation performance. These findings underscore the importance of the proposed reordering strategy, which ensures that the sequential fusion process begins with a valid and informative modality, thereby enabling more stable and effective feature integration.

## G.4 SENSITIVE ANALYSIS OF DIFFERENT BOTTLENECK SIZE

To evaluate the sensitivity of SMSN to the bottleneck dimensionality and the sequential fusion order, we conducted additional experiments using various combinations of bottleneck sizes (see Table 9). Since the channel dimension of each modality is 128, we tested a range of bottleneck sizes around this value. The results indicate that SMSN achieves the best performance when the two bottleneck dimensions are set to 128 and 256. This observation suggests that the chosen bottleneck sizes are appropriate, as excessively deviating from the original modality channel dimension may hinder effective feature transformation and cross-modal interaction.

## G.5 SEGMENTATION RESULTS ON BRATS24

Missing modality segmentation results of different fusion-based models on the BRATS24 dataset (200 samples) is illustrated in Table 10. Our proposed approach consistently outperforms other fusion-based methods across all missing modality configurations, demonstrating its superior generalization performance.

## G.6 MORPHOLOGICAL DIFFERENCE BETWEEN GLIOMAS AND METASTASES

As illustrated in Figure 5, gliomas typically originate within the brain parenchyma and present as infiltrative lesions with indistinct margins, often demonstrating heterogeneous signal intensities and irregular, infiltrative enhancement patterns. In contrast, metastases usually appear as well-circumscribed, round or ovoid lesions with sharp boundaries, often accompanied by pronounced peritumoral edema that exceeds the size of the enhancing lesion itself. These differences in margin definition, enhancement pattern, and surrounding edema provide important radiological clues for differential diagnosis.

Table 8: Missing modality segmentation results of SMSN on BRATS18 with fixed and random fusion order. Fixed 1 means fixed order and started from a modality contains less information (T1). The order is T1, T2, Flair and T1c. Fixed 2 means fixed order and started from a modality with more information (T1c). The order is T1c, T2, Flair and T1. $\sim$ means without. From top to bottom, it illustrates the results of WT, TC, and ET respectively.

| Method | T2 | T1C | T1 | F | T2 T1C | T1 T1C | F T1 | T1 T2 | F T2 | F T1C | $\sim$T2 | $\sim$T1C | $\sim$T1 | $\sim$F | - | AVG. |
|---|---|---|---|---|---|---|---|---|---|---|---|---|---|---|---|---|
| Fixed 1 | 83.49 | 71.72 | 73.72 | 86.21 | 85.13 | 77.12 | 86.94 | 84.88 | 87.70 | 88.01 | 87.52 | 87.88 | 88.24 | 85.44 | 88.29 | 84.16 |
| Fixed 2 | 83.00 | 71.79 | 72.93 | 87.44 | 85.18 | 77.32 | 89.04 | 85.37 | 89.42 | 89.56 | 89.14 | 89.10 | 90.09 | 85.46 | 89.95 | 84.99 |
| Random only | 81.53 | 70.11 | 69.02 | 86.44 | 84.14 | 75.69 | 88.00 | 84.51 | 88.53 | 87.96 | 87.54 | 88.49 | 88.15 | 85.12 | 88.34 | 83.69 |
| Random and present first | 85.49 | 76.33 | 76.26 | 84.38 | 86.45 | 79.62 | 87.82 | 86.61 | 88.51 | 88.71 | 89.09 | 89.98 | 86.93 | 89.39 | 88.71 | 85.62 |
| Fixed 1 | 58.51 | 76.79 | 62.39 | 60.29 | 79.64 | 79.45 | 68.44 | 68.73 | 65.12 | 79.31 | 80.89 | 69.38 | 79.27 | 81.33 | 81.06 | 72.70 |
| Fixed 2 | 63.98 | 77.26 | 59.98 | 62.89 | 76.67 | 79.90 | 69.03 | 69.01 | 68.81 | 78.80 | 80.46 | 70.41 | 79.58 | 80.48 | 80.54 | 73.19 |
| Random only | 55.38 | 76.30 | 63.09 | 64.04 | 77.98 | 77.90 | 69.81 | 66.95 | 68.15 | 77.63 | 79.54 | 70.61 | 78.73 | 78.70 | 78.99 | 72.15 |
| Random and present first | 68.27 | 79.72 | 62.34 | 61.85 | 83.57 | 80.75 | 67.70 | 68.55 | 70.78 | 82.35 | 83.17 | 70.77 | 82.81 | 82.96 | 82.35 | 75.20 |
| Fixed 1 | 41.56 | 67.62 | 33.80 | 33.53 | 73.79 | 72.06 | 39.26 | 45.16 | 41.44 | 73.27 | 73.85 | 44.66 | 72.88 | 74.22 | 73.71 | 57.39 |
| Fixed 2 | 41.51 | 73.46 | 34.65 | 32.12 | 72.66 | 74.91 | 38.13 | 42.92 | 42.70 | 75.93 | 77.52 | 42.65 | 76.20 | 75.84 | 75.61 | 58.45 |
| Random only | 31.29 | 63.77 | 33.55 | 33.36 | 65.90 | 64.68 | 39.02 | 39.87 | 40.52 | 66.80 | 65.82 | 42.53 | 69.51 | 66.34 | 67.40 | 52.58 |
| Random and present first | 44.51 | 78.87 | 36.55 | 39.31 | 79.70 | 79.37 | 44.42 | 46.25 | 46.09 | 78.58 | 80.30 | 45.71 | 78.91 | 78.70 | 78.58 | 62.39 |

## G.7 COMPARISON OF COMPUTATIONAL COSTS

Compared with pure parallel fusion frameworks such as mmFormer, SMSN does not require a substantial increase in the number of learnable parameters, as shown in Table 11. Despite a relatively longer training time, SMSN achieves superior segmentation accuracy while maintaining a comparable inference time. This demonstrates that sequential fusion can provide performance gains without incurring significant parameter overhead.

Table 9: Missing modality segmentation results of SMSN on BRATS18 with (two) different bottleneck size. $\sim$ means without. From top to bottom, it illustrates the results of WT, TC, and ET respectively.

| Size | T2 | T1C | T1 | F | T2 T1C | T1 T1C | F T1 | T1 T2 | F T2 | F T1C | $\sim$T2 | $\sim$T1C | $\sim$T1 | $\sim$F | - | AVG. |
|---|---|---|---|---|---|---|---|---|---|---|---|---|---|---|---|---|
| 64, 64 | 80.49 | 69.39 | 66.27 | 83.77 | 82.23 | 71.93 | 84.82 | 81.90 | 86.02 | 84.86 | 84.92 | 86.63 | 86.44 | 82.20 | 86.54 | 81.23 |
| 64, 128 | 82.64 | 69.30 | 70.92 | 86.84 | 84.15 | 76.01 | 88.62 | 84.08 | 88.42 | 88.14 | 88.35 | 88.91 | 89.02 | 84.55 | 89.07 | 83.93 |
| 128, 256 | 85.49 | 76.33 | 76.26 | 84.38 | 86.45 | 79.62 | 87.82 | 86.61 | 88.51 | 88.71 | 89.09 | 89.98 | 86.93 | 89.39 | 88.71 | 85.62 |
| 256, 512 | 82.38 | 71.40 | 73.46 | 86.91 | 84.70 | 77.03 | 88.67 | 84.94 | 88.60 | 87.89 | 88.32 | 88.91 | 89.05 | 85.28 | 89.40 | 84.46 |
| 512, 512 | 82.40 | 72.42 | 70.22 | 83.91 | 84.29 | 76.34 | 85.98 | 83.98 | 87.91 | 86.51 | 86.25 | 87.17 | 87.14 | 84.46 | 87.08 | 83.79 |
| 64, 64 | 59.64 | 76.34 | 58.58 | 59.63 | 78.39 | 77.12 | 67.74 | 67.03 | 65.07 | 78.69 | 80.02 | 69.74 | 79.28 | 79.77 | 80.36 | 71.82 |
| 64, 128 | 55.55 | 74.59 | 57.80 | 56.18 | 78.10 | 77.54 | 65.07 | 62.95 | 60.43 | 78.59 | 78.64 | 64.46 | 77.69 | 78.64 | 78.80 | 69.66 |
| 128, 256 | 68.27 | 79.72 | 62.34 | 61.85 | 83.57 | 80.75 | 67.70 | 68.55 | 70.78 | 82.35 | 83.17 | 70.77 | 82.81 | 82.96 | 82.35 | 75.20 |
| 256, 512 | 52.67 | 75.37 | 61.57 | 63.30 | 77.04 | 78.39 | 69.28 | 65.77 | 66.15 | 77.15 | 78.92 | 69.23 | 77.26 | 79.17 | 78.72 | 71.33 |
| 512, 512 | 57.87 | 74.77 | 54.90 | 58.13 | 76.57 | 75.10 | 66.05 | 63.24 | 63.87 | 75.99 | 77.53 | 67.92 | 76.81 | 77.51 | 78.12 | 69.33 |
| 64, 64 | 34.55 | 65.47 | 29.14 | 29.70 | 70.85 | 68.59 | 35.11 | 36.04 | 35.48 | 71.93 | 72.11 | 40.73 | 71.85 | 71.36 | 71.60 | 52.95 |
| 64, 128 | 35.97 | 63.77 | 29.56 | 27.60 | 69.45 | 65.99 | 32.53 | 39.40 | 37.03 | 68.82 | 69.10 | 38.47 | 71.52 | 69.68 | 73.31 | 53.64 |
| 128, 256 | 44.51 | 78.87 | 36.55 | 39.31 | 79.70 | 79.37 | 44.42 | 46.25 | 46.09 | 78.58 | 80.30 | 45.71 | 78.91 | 78.70 | 78.58 | 62.39 |
| 256, 512 | 39.47 | 71.99 | 35.51 | 34.90 | 72.85 | 72.08 | 40.57 | 44.47 | 43.62 | 73.89 | 73.94 | 46.15 | 73.81 | 73.83 | 73.50 | 58.04 |
| 512, 512 | 36.82 | 68.68 | 26.09 | 25.11 | 72.35 | 71.82 | 34.03 | 41.37 | 38.58 | 70.20 | 73.65 | 42.62 | 72.41 | 73.26 | 73.70 | 55.89 |

Table 10: Missing modality segmentation results of different fusion-based models trained on BRATS24. $\sim$ means without. From top to bottom, it illustrates the results of WT, TC, and ET respectively.

| Method | T2 | T1C | T1 | F | T2 T1C | T1 T1C | F T1 | T1 T2 | F T2 | F T1C | $\sim$T2 | $\sim$T1C | $\sim$T1 | $\sim$F | - | AVG. |
|---|---|---|---|---|---|---|---|---|---|---|---|---|---|---|---|---|
| mmFormer | 71.42 | 63.06 | 66.39 | 78.31 | 74.55 | 69.75 | 81.43 | 74.90 | 82.35 | 80.49 | 82.01 | 82.97 | 83.31 | 75.90 | 83.25 | 76.67 |
| M$^2$FTrans | 72.77 | 63.49 | 66.95 | 75.13 | 76.45 | 66.91 | 81.48 | 74.31 | 81.93 | 81.97 | 82.76 | 83.06 | 83.00 | 77.62 | 83.63 | 76.76 |
| MMMViT | 27.41 | 23.77 | 24.03 | 30.59 | 26.41 | 23.66 | 29.34 | 26.91 | 30.78 | 28.22 | 27.95 | 29.57 | 28.89 | 26.06 | 38.45 | 28.14 |
| IMS$^2$Trans | 63.81 | 53.80 | 57.04 | 69.49 | 69.51 | 59.56 | 76.10 | 70.24 | 76.56 | 74.74 | 75.70 | 77.40 | 78.35 | 71.54 | 78.47 | 70.15 |
| SMSN (ours) | 71.67 | 63.56 | 69.47 | 82.67 | 76.73 | 66.93 | 82.76 | 72.38 | 83.26 | 83.50 | 84.07 | 84.88 | 84.85 | 74.91 | 84.70 | 77.76 |
| mmFormer | 8.92 | 35.27 | 12.59 | 11.48 | 41.74 | 41.67 | 18.49 | 19.07 | 13.45 | 41.83 | 41.54 | 17.65 | 28.87 | 41.62 | 42.07 | 27.75 |
| M$^2$FTrans | 6.27 | 21.31 | 10.22 | 8.95 | 22.63 | 23.08 | 11.45 | 7.06 | 9.07 | 24.18 | 25.05 | 10.32 | 24.19 | 23.66 | 24.59 | 16.80 |
| MMMViT | 1.73 | 4.27 | 2.39 | 1.48 | 4.25 | 4.16 | 2.82 | 2.23 | 2.14 | 4.29 | 4.43 | 2.65 | 4.39 | 4.23 | 4.51 | 3.33 |
| IMS$^2$Trans | 5.08 | 28.67 | 10.50 | 7.90 | 32.13 | 36.90 | 11.95 | 9.46 | 7.91 | 33.13 | 36.65 | 13.65 | 33.97 | 38.96 | 38.46 | 23.02 |
| SMSN (ours) | 9.14 | 35.44 | 17.43 | 17.43 | 39.09 | 45.61 | 27.37 | 9.80 | 21.32 | 44.57 | 47.21 | 22.55 | 43.97 | 45.36 | 42.60 | 31.26 |
| mmFormer | 21.84 | 52.29 | 28.83 | 20.99 | 51.90 | 59.30 | 25.31 | 27.86 | 20.94 | 55.62 | 60.08 | 26.81 | 55.21 | 57.25 | 57.25 | 41.43 |
| M$^2$FTrans | 22.38 | 49.21 | 28.72 | 17.46 | 52.74 | 57.10 | 25.05 | 26.10 | 28.14 | 49.97 | 57.97 | 27.61 | 55.30 | 58.43 | 57.89 | 40.94 |
| MMMViT | 28.95 | 55.58 | 27.38 | 30.91 | 55.14 | 58.39 | 29.97 | 33.83 | 31.50 | 58.27 | 62.91 | 33.57 | 55.71 | 59.12 | 60.50 | 45.45 |
| IMS$^2$Trans | 20.13 | 28.41 | 9.87 | 9.22 | 37.16 | 38.94 | 14.08 | 17.26 | 24.92 | 35.56 | 43.92 | 22.02 | 42.64 | 44.17 | 49.98 | 29.22 |
| SMSN (ours) | 22.07 | 51.00 | 32.77 | 25.54 | 53.82 | 60.84 | 28.28 | 27.00 | 27.55 | 55.13 | 59.68 | 28.15 | 54.54 | 60.53 | 58.27 | 43.01 |

In addition, SMSN delivers strong performance under incomplete-modality scenarios without introducing notable computational overhead relative to attention-based fusion methods. In particular, compared with the second-best approach, M$^2$FTrans, the Two-Stage Information Bottleneck Fusion Module in SMSN adds only a marginal number of trainable parameters and results in nearly identical training time. These observations indicate that SMSN attains its improvements without imposing a substantial trade-off in computational efficiency when compared with attention-based fusion strategies.

Table 11: Comparison of the number of model parameters, Gflops, training time, inference time, throughput, complexity, and average segmentation performance of different models on different classes (WT/TC/ET). Assume there are n modalities and the feature dimension of each modality is d.

| | Parameters (M) | Gflops | Training Time/Epoch (s) | Inference Time/Epoch (s) | Throughput | Complexity | DSC on BRATS18 | DSC Generalization |
|---|---|---|---|---|---|---|---|---|
| mmFormer | 36.65 | 123.83 | 53.89 | 34.50 | 2.66 | $O(nd^2)$ | 84.88/73.25/54.34 | 28.43/8.69/5.78 |
| M$^2$FTrans | 13.49 | 113.36 | 110.11 | 28.25 | 1.27 | $O(n^2d)$ | 85.39/72.78/54.95 | 55.16/38.62/31.74 |
| MMMViT | 16.98 | 107.51 | 48.80 | 12.26 | 2.87 | $O(n^2d)$ | 78.54/71.39/55.21 | 49.96/36.50/30.64 |
| IMS$^2$Trans | 4.49 | 101.18 | 42.27 | 44.53 | 3.31 | $O(n^2d)$ | 83.37/68.76/54.56 | 54.98/39.04/30.64 |
| SMSN (ours) | 14.84 | 111.83 | 109.22 | 33.09 | 1.28 | $O(nd^2)$ | 85.62/75.20/62.39 | 57.03/45.70/36.89 |

# H LLM USAGE STATEMENT

We employed ChatGPT 5 to help polish the language and improve the readability. No LLMs were involved in designing experiments, analyzing data, or contributing to the scientific findings of this work.

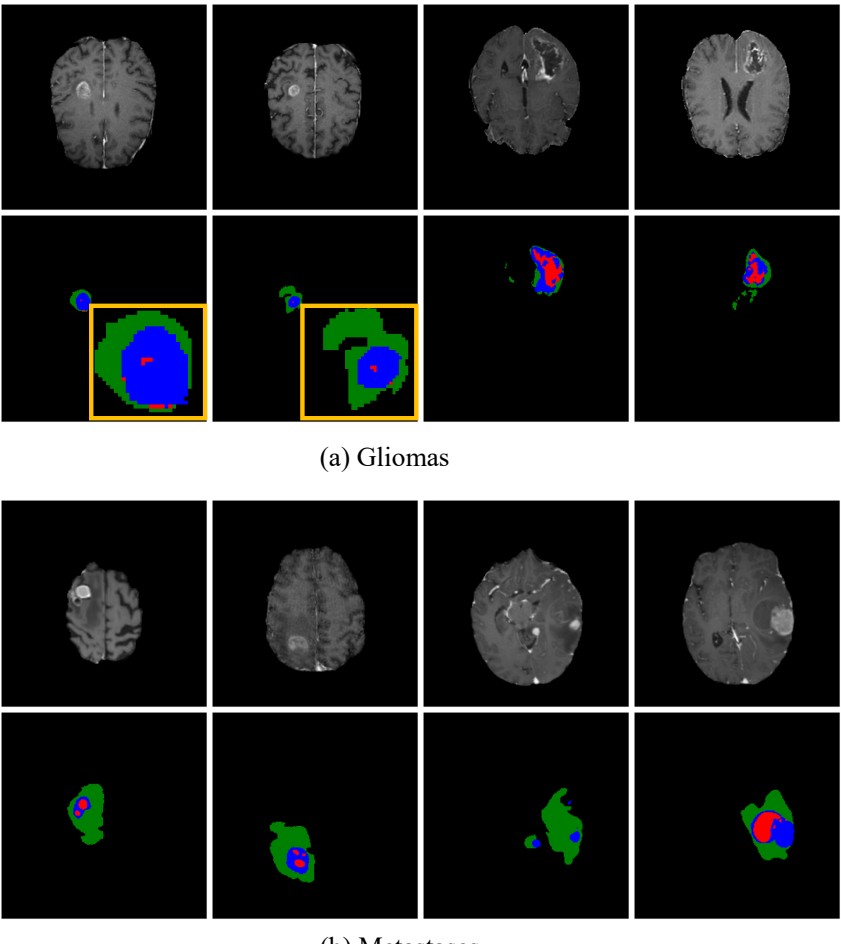

(a) Gliomas

(b) Metastases

Figure 5: Comparison between gliomas and metastases. Gliomas are typically infiltrative with ill-defined margins and heterogeneous enhancement, whereas metastases usually appear as well-circumscribed lesions with sharp boundaries and prominent peritumoral edema.

