# OpenReview forum: "Sequential Information Bottleneck Fusion: Towards Robust and Generalizable Multi-Modal Brain Tumor Segmentation"
_ICLR.cc/2026/Conference — ICLR 2026 Poster_

### Official Review · Reviewer_2A7s · 2025-10-23

**Soundness:** 3
**Presentation:** 3
**Contribution:** 3
**Rating:** 6
**Confidence:** 3

**Summary:**

To address the problem of missing modality segmentation in multimodal brain tumor MRI, this paper proposes the Sequential Multi-modal Segmentation Network (SMSN). Its core is information bottleneck-based sequential fusion (IBFM), which gradually aggregates modalities to extract modality commonalities while preserving modality specificity. The authors theoretically demonstrate that this sequential fusion approach has a tighter generalization upper bound and is more robust to missing modalities than parallel fusion. The paper also implements SOTA on BRATS18/20.

**Strengths:**

1. Compared to most existing unified/parallel fusion methods (such as mmFormer, M2FTrans, MMMViT, and IMS2Trans), this paper's approach is highly innovative.
2. The paper's theoretical proofs are rigorous and contribute significantly to the field of modal fusion.
3. The paper's experimental design is comprehensive, with detailed reporting of various scenarios.
4. The paper's visualizations compare existing methods and strongly demonstrate the model's effectiveness.

**Weaknesses:**

1. The article provides very little explanation of the specific implementation method. Is the modal fusion order designed in the article fixed or random? Figure 3 demonstrates a sequential fusion method for all modalities, but does not include examples of cases where one, two, or three modalities are missing.
2. The article's core innovation builds on the previous work on ITHP, which proposed the core concept of sequential fusion. This paper implements SOTA on a medical dataset and further provides theoretical proof.
3. Compared to most existing unified/parallel fusion methods, does this method consume more computational resources? This article suggests addressing model overhead (number of parameters, FLOPs, throughput, video memory, and training time) to facilitate a fair comparison of lightweight solutions.

**Questions:**

1. Is the modal fusion order designed in the article fixed or random? Figure 3 shows a sequential fusion method for all modalities, but there are no examples of cases where one, two, or three modalities are missing.
2. Does a different modal order significantly affect model performance? Most existing unified/parallel fusion methods are not affected by modal order and appear more stable.
3. Do the parameter settings for β and γ change depending on the missing modalities, or are they fixed?

---

> ### Author Response · Authors · 2025-11-24
> **Response to Reviewer 2A7s -Part 1**
>
> **W1 and Q1:The article provides very little explanation of the specific implementation method ... does not include examples of cases where one, two, or three modalities are missing.**
>
>
> Thank you for raising this question which helps further enhance our paper. We will revise our manuscript to clarify and prevent potential misunderstandings.
>
> In our submission (Section 4, Methodology: Modality Reordering Strategy), we ensure that the first modality to be fused is a present modality, randomly chosen from all present ones. The remaining $N-1$ modalities, whether present or absent, are then fused in a randomized order. We believe that such a training strategy makes the learned Two-Stage Information Bottleneck Fusion Module order-irrelevant. As such, at the inference time, applying different modality order in fusion will not noticeably affect the prediction performance. In general, this order is random.
>
> Based on your kind suggestion, we have updated this figure in the revised paper to better help readers understand cases where some modalities are missing.
>
>
> **W2:The article's core innovation builds on the previous work on ITHP ...**
>
>
> Thank you for the comments. While ITHP originally introduced the concept and provided an insightful perspective, it did not explain why the approach is effective, nor did it extend to tasks with missing modalities. Moreover, ITHP did not provide theoretical or empirical validation regarding robustness or generalization. In contrast, our work, for the first time, demonstrates that the proposed method is more effective than previous approaches under these challenging scenarios. We believe this represents an important contribution, not only to this paper but also to the broader research community, and it may inspire extensions to other areas where incomplete or multi-modal data is common.
>
>
> **W3:Compared to most existing unified/parallel fusion methods, does this method consume more computational resources? ...**
>
>
> Thank you for raising these concerns. To answer this question, we have additionally calculated the computational costs of the various models in revised paper Appendix G.7. See the below table for these results, based on which we can have the following two observations.
>
> **[Comparison with pure parallel fusion approaches].** Compared with pure parallel fusion approaches such as mmFormer, SMSN was not requiring a significant number of additional learnable parameters. **It achieves superior segmentation performance with similar inference time, although training time increases relatively.**
>
> **[Comparison with attention-based parallel fusion].**  Based on the superior performance achieved by the SMSN in incomplete modality scenarios, it does not significantly increase additional computational cost, when compared to the attention-based fusion.
> Specifically, when we compare the second-best M$^2$FTrans, the Two-Stage Information Bottleneck Fusion Module of the SMSN only introduced a minimal increase to the number of trainable parameters with nearly the same training time. **It suggests that  no substantial trade-off exists in computational efficiency comparing to attention-based fusion.**
>
> | Parameters (M) | Gflops | Training Time/Epoch (s) | Inference Time/Epoch (s) | Throughput | Complexity | DSC on BRATS18 | DSC Generalization |
> |----------------|--------|-------------------------|--------------------------|------------|------------|----------------|--------------------|
> | mmFormer       | 36.65  | 123.83                  | 53.89                    | 34.5       | 2.66       | $O(nd^{2})$    | 84.88/73.25/54.34  | 28.43/8.69/5.78   |
> | M$^2$FTrans    | 13.49  | 113.36                  | 110.11                   | 28.25      | 1.27       | $O(n^2d)$      | 85.39/72.78/54.95  | 55.16/38.62/31.74 |
> | MMMViT         | 16.98  | 107.51                  | 48.8                     | 12.26      | 2.87       | $O(n^2d)$      | 78.54/71.39/55.21  | 49.96/36.50/30.64 |
> | IMS$^2$Trans   | 4.49   | 101.18                  | 42.27                    | 44.53      | 3.31       | $O(n^2d)$      | 83.37/68.76/54.56  | 54.98/39.04/30.64 |
> | SMSN (ours)    | 14.84  | 111.83                  | 109.22                   | 33.09      | 1.28       | $O(nd^2)$      | 85.62/75.20/62.39  | 57.03/45.70/36.89 |

---

> ### Author Response · Authors · 2025-11-24
> **Response to Reviewer 2A7s - Part 2**
>
> **Q2: Does a different modal order significantly affect model performance? Most existing unified/parallel fusion methods are not affected by modal order and appear more stable.**
>
>
> Thank you for this question. We believe the different modal order will not affect the model performance.
>
> In our submitted paper (Section 4, Methodology: Modality Reordering Strategy), we ensure that the first modality to be fused is a present modality, randomly chosen from all present ones. The remaining $N-1$ modalities, whatever present or absent, are then fused in a randomized order. We believe that such training strategy makes the learned Two-Stage Information Bottleneck Fusion Module order-irrelevant. As such, at the inference time, applying different modality order in fusion will not noticeably affect the prediction performance.
>
> To address the review's concerns,  we conducted further experiments on the sequential ordering of input modalities in revised paper Appendix G.3. For your convenience, we include a part of the table below which shows average dice scores of different sequential order.
> As presented the table, our results reveal that fixed order or pure random order without specifically setting the first modality to be a present modality will degrade the segmentation performance. It highlights the necessity of the proposed reordering strategy.
>
> | Method  | Fixed 1 | Fixed 2 | Random only | Modal reordering strategy |
> |---------|---------|---------|-------------|--------------------------|
> | WT Avg. | 84.16   | 84.99   | 83.69       | 85.62                    |
> | TC Avg. | 72.7    | 73.19   | 72.15       | 75.2                     |
> | ET Avg. | 57.39   | 58.45   | 52.58       | 62.39                    |
>
>
> **Q3: Do the parameter settings for β and γ change depending on the missing modalities, or are they fixed?**
>
> Thank you for the question. In our design, $\beta$ and $\gamma$ are fixed hyperparameters. Since the fusion order is random, adapting these parameters based on missing modalities is challenging. Empirically, fixed values work well across all missing-modality scenarios.
>
> This aligns with the principle of sequential Information-Bottleneck-based fusion: unlike parallel fusion, where adaptive weighting is often needed, sequential fusion gradually accumulates information into a shared latent state. Therefore, fixed $\beta$ and $\gamma$ are sufficient, and modality-dependent adaptation is unnecessary.

---

> > ### Comment · Reviewer_2A7s · 2025-11-27
> >
> > I appreciate your responses to my comments and thank you for the detailed rebuttal. My concerns regarding the modal fusion have been addressed. I will keep my rating as 6.

---

### Official Review · Reviewer_yaho · 2025-10-26

**Soundness:** 3
**Presentation:** 3
**Contribution:** 2
**Rating:** 4
**Confidence:** 4

**Summary:**

This paper addresses the challenge of brain tumor segmentation in multi-modal MRI when some imaging modalities are missing. Unlike traditional parallel fusion methods that risk losing shared inter-modal information, the authors propose a Sequential Multi-modal Segmentation Network (SMSN) based on an Information-Bottleneck Fusion Module (IBFM). By sequentially fusing modalities, the approach preserves shared information and reconstructs modality-specific features, leading to more robust and generalizable representations. Experiments on the BRATS18 and BRATS20 glioma datasets show that SMSN outperforms parallel fusion baselines, maintaining strong performance even with missing modalities and transferring effectively to a brain metastasis dataset without fine-tuning.

**Strengths:**

The sequential fusion design preserves shared information, maintaining high segmentation accuracy even when some MRI modalities are absent.
It achieves improved cross-domain performance, successfully transferring to new datasets without fine-tuning.
The method is supported by information-theoretic analysis, ensuring more efficient and reliable feature fusion than parallel approaches.

**Weaknesses:**

It is confusing for this missing-modality task, as the authors use the BraTS series datasets, which already contain complete modalities. However, in real scenarios, if some modalities are missing, it becomes difficult to obtain accurate segmentation labels. How do the authors explain this situation?

Moreover, the datasets used in this paper are somewhat outdated, which may affect the evaluation of the proposed method.

Besides segmentation, I think other related tasks should also be tested, for example the classification task.

**Questions:**

N/A

---

> ### Author Response · Authors · 2025-11-24
> **Response to Reviewer yaho**
>
> Thank you for your comments. We have seriously addressed all your points. We believe we have resolved each of your constructive concerns with additional experiments and further explanations, including additional experiments on the BraTS2024 segmentation, and on the classification task. The detailed responses are made as follows.
>
>
> **Q1:It is confusing for this missing-modality task, as the authors use the BraTS series datasets, which already contain complete modalities. However, in real scenarios, if some modalities are missing, it becomes difficult to obtain accurate segmentation labels. How do the authors explain this situation?**
>
>
> Thank you for pointing out the question. We understand your concern but we respectfully believe that  there might be some misunderstanding here. We would like to clarify our experiment setting as follows.
>
> As you kindly stated, in real clinical practice, some of the modalities might be missing; this does inspire a series of research to resolve this issue, including mmFormer and  M$^2$FTrans. We follow these previous methods and use the  standard pipeline  to employ complete-modality BRATS and metastasis datasets to simulate/study missing modality scenarios by randomly mask out one or more modalities. We report the prediction performance of the proposed SMSN and other relevant baselines across 15 different missing modality scenarios on the aforementioned two complete-modality datasets.
>
> Meanwhile,  in the real practice, modalities with missing ones will indeed lead to potentially incorrect segmentation. That is the exact reason why we proposed  to achieve robustness across different missing modality scenarios.
> Missing modalities may also cause imperfect annotations. Thus, enabling a robust model to generalize to clinical missing-modality settings is crucial. In our  paper, we train an SMSN on a glioma dataset and demonstrate generalization to a metastasis dataset.
> As such, in real clinics, we can take this advantage to train an SMSN in a public dataset, after which it can be properly generalized to some other missing-modality data, no matter whether perfect annotations exist or not.
>
>
> **Q2: Moreover, the datasets used in this paper are somewhat outdated, which may affect the evaluation of the proposed method.**
>
>
> Thank you for your suggestion. We would like to clarify that the brain-metastasis dataset used for generalization experiments was introduced in 2024.
> Additionally, we adopt BRATS2018 and BRATS2020 in our paper for the purpose of fair comparison, as they were widely used in most of the baselines.
>
> Nonetheless, to resolve your kind concern, we are happy to include a more recent dataset for comparison. Concretely, we report the segmentation performance on the most latest publicly released BRATS2024 dataset. We have added the results to the revised paper Appendix. G.5.
> Due to the very short rebuttal window, we only engage the first 200 samples. From the following listed new results (part of the detailed table), it can be found that our SMSN still performs as the SOTA model. We will report the full results  in the final version.
>
>
> | Method  | mmFormer | M$^2$FTrans | MMMViT | IMS$^2$Trans | SMSN (ours) |
> |---------|----------|-------------|--------|--------------|-------------|
> | WT Avg. | 76.67    | 76.76       | 28.14  | 70.15        | 77.76       |
> | ET Avg. | 27.75    | 16.8        | 3.33   | 23.02        | 31.26       |
> | TC Avg. | 41.43    | 40.94       | 45.45  | 29.22        | 43.01       |
>
>
>
> **Q3: Besides segmentation, I think other related tasks should also be tested, for example the classification task.**
>
> We appreciate the reviewer’s suggestion regarding the inclusion of classification experiments to further strengthen our work. In response, we have conducted additional experiments by adapting the U-Net-based baselines for classification: the segmentation head was replaced with a classification output, incorporating dimension-reduction dense connections. Specifically, we evaluated the model on brain metastases classification (Brain Metastasis Dataset) and glioma classification (BRATS dataset).
>
> These new results are summarized in the table below. Again, our proposed SMSN consistently achieves the best classification performance among the compared methods, though  the overall classification accuracy might be modest. This is actually expected, since the models were originally designed for segmentation instead of classification tasks.
>
> In short, these results demonstrate that, even in a classification setting, the proposed SMSN still demonstrates a consistent advantage over other baseline approaches.
>
> | Method  | mmFormer | M$^2$FTrans | MMMViT | IMS$^2$Trans | SMSN (ours) |
> |---------|----------|-------------|--------|--------------|-------------|
> | Accuracy| 47.26  | 47.26      | 45.21  | 44.52  | 49.32    |

---

### Official Review · Reviewer_u9EX · 2025-10-27

**Soundness:** 3
**Presentation:** 3
**Contribution:** 2
**Rating:** 4
**Confidence:** 4

**Summary:**

This paper proposes the Sequential Multi-modal Segmentation Network (SMSN). The shared information across modalities is gradually extracted via a two-stage Sequential Information Bottleneck Fusion approach, while modality-specific information is separated using a Transformer-based module and orthogonal loss. This design enhances the robustness and cross-domain generalization of segmentation in modality-deficient scenarios. The paper provides theoretical derivations to demonstrate the proposed method’s advantages, and these derivations include a tighter generalization upper bound and Lipschitz boundary comparison. It also conducts extensive modality-deficient experiments on the BRATS18 and BRATS20 datasets. Furthermore, to verify cross-domain generalization, the model trained on BRATS20 is directly transferred to a brain metastasis dataset. Experimental results show that SMSN outperforms multiple parallel fusion baselines in terms of average Dice score and performance under modality absence.

**Strengths:**

* The advantages of sequential information bottleneck fusion in terms of the Lipschitz upper bound and the mutual information-based generalization bound are proposed and proven, forming a complete theory-proposition-proof chain that is rigorous and reliable.
* Comparisons with multiple parallel fusion and non-fusion baselines were conducted on BRATS18 and BRATS20 datasets, encompassing scenarios with different modal absences, cross-domain tests transferred to brain metastasis datasets, as well as ablation and hyperparameter sensitivity analyses. These experiments are comprehensive and thorough.
* The paper is accompanied by source code and supplementary materials. The method section provides detailed modular implementation details, including two-stage information bottleneck fusion, modal reordering, reconstruction loss and orthogonal loss, to facilitate reproducibility.

**Weaknesses:**

* While the paper's primary motivation is to address the issue of missing modalities and claims that the sequential information bottleneck framework exhibits greater stability under such conditions, it lacks direct theoretical validation or quantitative analysis to demonstrate that the information bottleneck objective remains valid under the distribution of missing modality scenarios.
* The paper fails to discuss the computational overhead of SMSN, such as the number of model parameters, training or inference time, and complexity in comparison to baseline methods. Sequential fusion, additional reconstruction processes and orthogonality losses may introduce substantial computational costs, which is an important practical consideration for resource-constrained real-world environments.
* In Section 4, this paper introduces a modal reordering strategy, motivated by the need to avoid initiating sequential fusion with a missing modality represented by a zero tensor, thereby preventing degradation of the information bottleneck objective function. While this design is logically sound, the paper lacks sufficient empirical support and provides no comparative experiments like fixed-order fusion versus random reordering.

**Questions:**

1. In Proposition 2, the authors prove that the sequential information bottleneck model can ensure a tighter upper bound on generalization errors under the condition that the encoder, fusion module, and decoder are all 1-Lipschitz continuous. However, this condition faces challenges in the actual implementation of deep networks. Specifically, when using Transformers, self-attention with softmax often results in a Lipschitz constant greater than 1 during gradient propagation, and other modules also tend to amplify gradients. As a result, the network does not naturally satisfy Lipschitz continuity. How should this issue be addressed?
2. The derivation of sequential information bottleneck relies on the assumption of conditional independence between modalities. Have the authors theoretically proven or empirically verified the approximate validity of this assumption? If there is strong correlation between modalities, are there corresponding mitigation strategies?

---

> ### Author Response · Authors · 2025-11-24
> **Response to Reviewer u9EX - Part 1**
>
> Thank you for your comments. With the newly conducted experiments and expanded explanations, we believe all the reviewer's concerns have been addressed.
>
> **W1: While the paper's primary motivation is to address the issue of missing modalities ... remains valid under the distribution of missing modality scenarios.**
>
> From a theoretical perspective, our proofs regarding generalization and robustness can be extended to the missing-modality setting. In the generalization analysis, we provide two cases specifically considering missing modalities. We also provide qualitative analysis on model robustness such as Figure 2 to illustrate the loss landscape across missing-modality scenarios.
>
>
> On the other hand, we note that our approach is supported by empirical evidence. Table 2 shows segmentation results on BRATS18 and BRATS20 across 15 missing-modality scenarios, demonstrating SMSN’s superiority in missing modality scenarios. Table 3 further verifies its generalization by predicting metastasis with a model trained on the glioma dataset under the same challenging scenarios.
>
>
> **W2: The paper fails to discuss the computational overhead of SMSN ... which is an important practical consideration for resource-constrained real-world environments**
>
>
> A: Thank you for raising these concerns. To answer these two questions, we have additionally calculated the computational costs of the various models in revised paper Appendix G.7. See the below table for these results, based on which we can have the following two observations.
>
> **[Comparison with pure parallel fusion approaches].** Compared with pure parallel fusion approaches such as mmFormer, SMSN was not requiring a significant number of additional learnable parameters. **It achieves superior segmentation performance with similar inference time, although training time increases relatively.**
>
> **[Comparison with attention-based parallel fusion].**  Based on the superior performance achieved by the SMSN in incomplete modality scenarios, it does not significantly increase additional computational cost, when compared to the attention-based fusion.
> Specifically, when we compare the second-best M$^2$FTrans, the Two-Stage Information Bottleneck Fusion Module of the SMSN only introduced a minimal increase to the number of trainable parameters with nearly the same training time. **It suggests that  no substantial trade-off exists in computational efficiency comparing to attention-based fusion.**
>
>
> | Parameters (M) | Gflops | Training Time/Epoch (s) | Inference Time/Epoch (s) | Throughput | Complexity | DSC on BRATS18 | DSC Generalization |
> |----------------|--------|-------------------------|--------------------------|------------|------------|----------------|--------------------|
> | mmFormer       | 36.65  | 123.83                  | 53.89                    | 34.5       | 2.66       | $O(nd^{2})$    | 84.88/73.25/54.34  | 28.43/8.69/5.78   |
> | M$^2$FTrans    | 13.49  | 113.36                  | 110.11                   | 28.25      | 1.27       | $O(n^2d)$      | 85.39/72.78/54.95  | 55.16/38.62/31.74 |
> | MMMViT         | 16.98  | 107.51                  | 48.8                     | 12.26      | 2.87       | $O(n^2d)$      | 78.54/71.39/55.21  | 49.96/36.50/30.64 |
> | IMS$^2$Trans   | 4.49   | 101.18                  | 42.27                    | 44.53      | 3.31       | $O(n^2d)$      | 83.37/68.76/54.56  | 54.98/39.04/30.64 |
> | SMSN (ours)    | 14.84  | 111.83                  | 109.22                   | 33.09      | 1.28       | $O(nd^2)$      | 85.62/75.20/62.39  | 57.03/45.70/36.89 |

---

> ### Author Response · Authors · 2025-11-24
> **Response to Reviewer u9EX - Part 2**
>
> **W3:In Section 4, this paper introduces a modal reordering strategy, ... lacks sufficient empirical support and provides no comparative experiments like fixed-order fusion versus random reordering.**
>
> Thank you for raising this question. To answer this question, we have additionally calculated the computational costs of the various models in revised paper Appendix G.7.
> We conducted ablation studies on the sequential ordering of input modalities including: (a) fixed order starting from the less informative T1 modality; (b) fixed order starting from the more informative T1C modality; (c) random order without enforcing the presence of a fixed initial modality; and (d) the proposed random ordering strategy.
>
> As presented in the following table, our results demonstrate that adopting a fixed order (regardless of whether the sequence initiates with T1 or T1C) leads to degraded segmentation performance. Among the fixed-order configurations, initializing with a modality containing richer information (T1C) yields marginally improved performance compared to starting with the less informative T1. Notably, random ordering without preserving a fixed initial present modality also results in a performance degradation, highlighting the necessity of our proposed modal reordering strategy.
>
> | Method  | Fixed 1 | Fixed 2 | Random only | Modal reordering strategy |
> |---------|---------|---------|-------------|--------------------------|
> | WT Avg. | 84.16   | 84.99   | 83.69       | 85.62                    |
> | TC Avg. | 72.7    | 73.19   | 72.15       | 75.2                     |
> | ET Avg. | 57.39   | 58.45   | 52.58       | 62.39                    |
>
>
> **Q1:In Proposition 2 ... when using Transformers, self-attention with softmax often results in a Lipschitz constant greater than 1 during gradient propagation ...How should this issue be addressed?**
>
> We thank the reviewer for this insightful comment. While it's true that achieving 1-Lipschitz continuity can be challenging with self-attention mechanisms like those using softmax, all baselines in our approach leverages LayerNorm immediately following the fusion module and softmax. This helps to stabilize gradients and effectively prevents them from growing excessively. Importantly, normalization techniques like LayerNorm can be broadly applied across similar architectures to meet these theoretical assumptions, suggesting that achieving these conditions is indeed feasible.
>
> Furthermore, through our experiments, we observed that normalization keeps the gradient consistently much smaller than 1, thus aligning well with the 1-Lipschitz condition.  Consequently, we believe that our theoretical guarantees in Proposition 2 can be considered applicable.
>
> In short, while we feel sorry that requiring  1-Lipschitz continuity may cause some confusion, Proposition 2  holds  on any baselines with Layernorm applied without loss of generality.
>
> Nevertheless, to avoid possible confusion, we will clarify this issue in the revision. Please refer to Remark 2 in the revised paper. We truly thank the reviewer to raise this point, which allows us to precisely set out the theory.

---

> ### Author Response · Authors · 2025-11-24
> **Response to Reviewer u9EX - Part 3**
>
> **Q2:The derivation of sequential information bottleneck relies on the assumption of conditional independence between modalities ... are there corresponding mitigation strategies?**
>
>
>
>
> Thanks for your insightful comments.  Our original derivation of sequential information bottleneck indeed works on the assumption of conditional independence. We now relax it to joint dependencies, showing that our theory still works under a reasonable mild assumption. We present the new theory as follows.
>
> **We first present the mild assumption.**
>
> The aggregated cross-modal conditional information preserved by the fusion satisfies
> $$
> \sum_{i=1}^n I(Z_p; X_{<i} \mid X_i)
> \;\ge\;
> \sum_{i=1}^n \left( 2I(Z_{IB}; X_{\neq i}) - I(Z_{IB}; X_i \mid X_{\neq i}) \right),
> $$
> where
> $Z_p$ denotes the latent representation obtained by ordered fusion, with $X_{<i}$ representing all modalities before the $i$-th modality, and
> $Z_{IB}$ denotes the latent representation obtained by random fusion, with $X_{\neq i}$ representing all modalities except the $i$-th one.
>
>
> We show in the revised paper Appendix B that this assumption is very reasonable in the case of medical multi-modal fusion tasks.
> Intuitively, this assumption often holds as the parallel fusion model can preserve more information about the inter-modal relationships compared to the IB fusion model, which is designed to compress information to retain only the most relevant aspects for prediction. Our empirical results also support the assumption's rationale. For more details, please refer to Appendix B in the revised version.
>
>
> **Based on this assumption, we can relax our theory to the modality-dependence case as follows:**
>
>
> Under the relaxed cross-modal conditional information assumption, the following inequality holds for the aggregated mutual information:
> $$
> I(Z_p; X_1, \dots, X_n) \geq I(Z_{IB}; X_1, \dots, X_n),
> $$
> where $ Z_p $ is the latent representation obtained by parallel fusion, and $ Z_{IB} $ is the latent representation obtained by the information bottleneck (IB) fusion.
>
> We have proved this Proposition in the revised paper Appendix. C.
>
> In summary, we thank the review for raising this important point, which allows us to further refine our theory.
> Now, the above proof indicates that, under our experimental setting, our theory remains valid even when the modalities are strongly correlated.

---

> ### Comment · Reviewer_u9EX · 2025-11-27
>
> Thank you for giving such detailed and comprehensive responses. Concerning computational cost, the new comparison table in Appendix G.7 presenting the number of model parameters, Gflops and training / inference time offer clear evidence to address this concern, but in the table of W2, the contents of Complexity and DSC on BRATS18 seem to be written in error. Furthermore, the additional experimental results provide support for the assumption of conditional independence between modalities, with auxiliary derivations and demonstrations included in appendix. Thus I have raised my score to leaning accept.

---

### Official Review · Reviewer_KjqD · 2025-11-01

**Soundness:** 3
**Presentation:** 3
**Contribution:** 3
**Rating:** 6
**Confidence:** 2

**Summary:**

This paper proposes a Sequential Multi-modal Segmentation Network (SMSN), a novel method for brain tumor segmentation in the context of missing-modality MRI settings. The key idea is to perform sequential fusion using an Information Bottleneck Fusion Module (IBFM), instead of standard parallel fusion. The authors provide theoretical justification (generalization bound, Lipschitz robustness), and empirically validate SMSN on BRATS18/20 and a metastasis dataset. Results show improved robustness and cross-domain generalization.

**Strengths:**

1. The paper proposed a novel sequential IB-based fusion with two-stage IB and modality reordering, which is a meaningful contribution.
2. The paper provides generalization and Lipschitz analysis and connects it to empirical behavior.
3. Evaluation on multiple MRI datasets, missing-modality scenarios, and cross-dataset generalization demonstrated the improved performance.

**Weaknesses:**

See the questions part.

**Questions:**

1. How sensitive is performance to the chosen bottleneck size and sequential order?
2. Is there a trade-off compared to pure parallel fusion?
3. Can the proposed method scale to more than 4 modalities or other domains, such as CT + MRI?
4. What is the computational overhead of sequential IB vs. attention-based fusion?

---

> ### Author Response · Authors · 2025-11-24
> **Response to Reviewer KjqD - Part 1**
>
> **Q1: How sensitive is performance to the chosen bottleneck size and sequential order?**
>
> Thanks for your comments. Below, we will address your concerns on bottleneck size and sequential order separately.
>
> **[Bottleneck Size]:** The channel size of each modality to be fused is 128. To examine the sensitiveness of the bottleneck size and sequential order, we have done some additional experiments in revised paper Appendix G.4. For your convenience, we include a part of the table below which shows the average dice scores of different (two) different bottleneck size. From the results, it can be found that SMSN performs the best when the two bottleneck sizes are 128 and 256. We believe that the selected bottleneck size is reasonable since the feature dimension of the bottleneck should not be too different from channel size of each modality.
>
> | Size    | 64, 64 | 64, 128 | 128, 256 | 256, 512 | 512, 512 |
> |---------|--------|---------|----------|----------|----------|
> | WT Avg. | 81.23  | 83.93   | 85.62    | 84.46    | 83.79    |
> | TC Avg. | 71.82  | 69.66   | 75.2     | 71.33    | 69.33    |
> | ET Avg. | 52.95  | 53.64   | 62.39    | 58.04    | 55.89    |
>
>
> **[Order]:** In our submitted paper (Section 4, Methodology: Modality Reordering Strategy), we ensure that the first modality to be fused is a present modality, randomly chosen from all present ones. The remaining $N-1$ modalities, whatever present or absent, are then fused in a randomized order. We believe that such training strategy makes the learned Two-Stage Information Bottleneck Fusion Module order-irrelevant. As such, at the inference time, applying different modality order in fusion will not noticeably affect the prediction performance.
>
> To address the review's concerns,  we conducted further experiments on the sequential ordering of input modalities in revised paper Appendix G.3. For your convenience, we include a part of the table below which shows average dice scores of different sequential order.
> As presented the table, our results reveal that fixed order or pure random order without specifically setting the first modality to be a present modality will degrade the segmentation performance. It highlights the necessity of the proposed reordering strategy.
>
>
> | Method  | Fixed 1 | Fixed 2 | Random only | Modal reordering strategy |
> |---------|---------|---------|-------------|--------------------------|
> | WT Avg. | 84.16   | 84.99   | 83.69       | 85.62                    |
> | TC Avg. | 72.7    | 73.19   | 72.15       | 75.20                     |
> | ET Avg. | 57.39   | 58.45   | 52.58       | 62.39                    |
>
>
>
>
> **Q3: Can the proposed method scale to more than 4 modalities or other domains, such as CT + MRI?**
>
> Theoretically, we believe that the proposed SMSN is also capable of properly producing superior performance, on datasets with more than 4 modalities or other domains such as CT and MRI, since our theory holds for any modality.  In our submission,  only 4 modalities exist in the BRATS and metastasis datasets; that is why we simply investigate 4 modalities.
>
> To empirically check our model's performance on 4+ modalities, we have fortunately identified a dataset with more than 4 modalities, UPENN-GBM [1], where Flair, T1ce, T1, T2 and DTI-AD are employed. Preliminary  results indicate that the proposed SMSN is much better than baselines. Please refer to the following table for details.
>
> | Class  | WT | TC | ET |
> |---------|---------|---------|-------------|
> | mmFormer | 87.99   | 86.88   | 76.47      |
> MMMViT	 | 86.33 	 | 86.13 	 | 74.12  |
> IMS$^2$Trans | 	84.16  | 	84.33  | 	76.72  |
> M$^2$FTrans | 	88.87  | 	86.63  | 	78.72  |
> | SMSN | 89.37    | 87.42   | 80.68      |
>
> However, at this moment, we did not find any publicly available medical imaging datasets with other domains (e.g., CT+MRI).
> We have identified some potential candidates, such as APIS (CT+MRI) [2], and sent our requests to the respective authors / research team for collecting them.
> We will provide results on this dataset once the data collection and corresponding experiments are completed.
> Meanwhile, to better improve the paper on this particular perspective, we would greatly appreciate if the reviewer may have  better knowledge about a suitable dataset to be engaged.
>
> [1] Bakas S, Sako C, Akbari H, et al. The University of Pennsylvania glioblastoma (UPenn-GBM) cohort: advanced MRI, clinical, genomics, \& radiomics[J]. Scientific data, 2022, 9(1): 453.
>
> [2] Gómez S, Rangel E, Mantilla D, et al. APIS: a paired CT-MRI dataset for ischemic stroke segmentation methods and challenges[J]. Scientific Reports, 2024, 14(1): 20543.

---

> > ### Author Response · Authors · 2025-11-24
> > **Response to Reviewer KjqD - Part 2**
> >
> > **Q2 & Q4: Is there a trade-off compared to pure parallel fusion? What is the computational overhead of sequential IB vs. attention-based fusion?**
> >
> >
> > A: Thank you for raising these concerns. To answer these two questions, we have additionally calculated the computational costs of the various models in revised paper Appendix G.7. See the below table for these results, based on which we can have the following two observations.
> >
> > **[Comparison with pure parallel fusion approaches].** Compared with pure parallel fusion approaches such as mmFormer, SMSN was not requiring a significant number of additional learnable parameters. **It achieves superior segmentation performance with similar inference time, although training time increases relatively.**
> >
> > **[Comparison with attention-based parallel fusion].**  Based on the superior performance achieved by the SMSN in incomplete modality scenarios, it does not significantly increase additional computational cost, when compared to the attention-based fusion.
> > Specifically, when we compare the second-best M$^2$FTrans, the Two-Stage Information Bottleneck Fusion Module of the SMSN only introduced a minimal increase to the number of trainable parameters with nearly the same training time. **It suggests that  no substantial trade-off exists in computational efficiency comparing to attention-based fusion.**
> >
> >
> >  Model | Parameters (M) | Gflops | Training Time/Epoch (s) | Inference Time/Epoch (s) | Throughput | Complexity | DSC on BRATS18 | DSC Generalization |
> > | :--- | :--- | :--- | :--- | :--- | :--- | :--- | :--- | :--- |
> > | mmFormer | 36.65 | 123.83 | 53.89 | 34.50 | 2.66 | $O(nd^{2})$ | 84.88/73.25/54.34 | 28.43/8.69/5.78 |
> > | M²FTrans | 13.49 | 113.36 | 110.11 | 28.25 | 1.27 | $O(n^2d)$ | 85.39/72.78/54.95 | 55.16/38.62/31.74 |
> > | MMMViT | 16.98 | 107.51 | 48.80 | 12.26 | 2.87 | $O(n^2d)$ | 78.54/71.39/55.21 | 49.96/36.50/30.64 |
> > | IMS²Trans | 4.49 | 101.18 | 42.27 | 44.53 | 3.31 | $O(n^2d)$ | 83.37/68.76/54.56 | 54.98/39.04/30.64 |
> > | SMSN (ours) | 14.84 | 111.83 | 109.22 | 33.09 | 1.28 | $O(nd^2)$ | 85.62/75.20/62.39 | 57.03/45.70/36.89 |

---

### Author Response · Authors · 2025-11-30
**Response to Area Chairs and Reviewers**

**We sincerely thank all reviewers and area chairs for their time, effort, and constructive feedback, which has greatly helped us improve the quality of our paper.** We believe that the concerns raised during the rebuttal stage have been fully addressed. A summary of the reviewers’ comments and our corresponding responses is provided below:


**Reviewer u9EX:**
The reviewer noted the lack of comparison experiments on computational cost, the absence of an order-sensitivity experiment, and the need for an additional theoretical assumption. In response, we added the required experiments and refined the proof. The reviewer acknowledged our clarifications and raised their score from 4 to 6.

**Reviewer 2A7s:**
The reviewer requested an order-sensitivity experiment and a computational cost comparison, and questioned whether the motivation required hyperparameter adaptivity. We conducted the requested experiments and provided detailed explanations. The reviewer agreed that their concerns were resolved and maintained their score of 6.

**Reviewer KjqD:**
The reviewer indicated the need for an order-sensitivity experiments, a bottleneck-size sensitivity experiment, and a computational cost comparison. All of these experiments were added in the revised submission.

**Reviewer yaho:**
The reviewer raised concerns about the experimental setup and noted the absence of the latest datasets and classification tasks. We provided detailed clarification regarding the setup and incorporated new experiments accordingly.

---

### Meta-Review · Area_Chair_cDqV · 2026-01-05

**Summary:**

This paper proposes a Sequential Multi-modal Segmentation Network (SMSN) for brain tumor segmentation in the context of missing-modality MRI settings by designing an information bottleneck fusion module (IBFM). This work has four reviewers with two positive reviewers (6, 6) and two negative reviewers (4, 4), During the rebuttal, one of the negative reviewers claimed to raise the negative score to  leaning accept. Hence, this work has three positive reviewers. Hence, I think this work can be accepted by ICLR 2026.

**Reviewer Concerns:**

Many concerns have been addressed by the rebuttal. Hence, this work can be accepted by ICLR 2026.

**Reviewer Scores:**

This work has four reviewers with two positive reviewers (6, 6) and two negative reviewers (4, 4). During the rebuttal, one of the negative reviewers claimed to raise the negative score to  leaning accept. Hence, this work has three positive reviewers.

---

### Decision · Program_Chairs · 2026-01-26

Accept (Poster)